# A proteomic atlas of insulin signalling reveals tissue-specific mechanisms of longevity assurance

Luke S Tain[1], Robert Sehlke[1,2], Chirag Jain[1], Manopriya Chokkalingam[2], Nagarjuna Nagaraj[3], Paul Essers[1], Mark Rassner[1], Sebastian Grönke[1], Jenny Froelich[1], Christoph Dieterich[4,5], Matthias Mann[3] (iD), Nazif Alic[6], Andreas Beyer[2,7,*] (iD) & Linda Partridge[1,6,**] (iD)

## Abstract

Lowered activity of the insulin/IGF signalling (IIS) network can ameliorate the effects of ageing in laboratory animals and, possibly, humans. Although transcriptome remodelling in long-lived IIS mutants has been extensively documented, the causal mechanisms contributing to extended lifespan, particularly in specific tissues, remain unclear. We have characterized the proteomes of four key insulin-sensitive tissues in a long-lived *Drosophila* IIS mutant and control, and detected 44% of the predicted proteome (6,085 proteins). Expression of ribosome-associated proteins in the fat body was reduced in the mutant, with a corresponding, tissue-specific reduction in translation. Expression of mitochondrial electron transport chain proteins in fat body was increased, leading to increased respiration, which was necessary for IIS-mediated lifespan extension, and alone sufficient to mediate it. Proteasomal subunits showed altered expression in IIS mutant gut, and gut-specific over-expression of the RPN6 proteasomal subunit, was sufficient to increase proteasomal activity and extend lifespan, whilst inhibition of proteasome activity abolished IIS-mediated longevity. Our study thus uncovered strikingly tissue-specific responses of cellular processes to lowered IIS acting in concert to ameliorate ageing.

**Keywords** ageing; insulin/IGF; mitochondria; proteasome; proteome
**Subject Categories** Ageing; Post-translational Modifications, Proteolysis & Proteomics; Signal Transduction
**Mol Syst Biol. (2017) 13: 939**

## Introduction

Reduced activity of the highly evolutionarily conserved insulin/IGF-like signalling (IIS) network in laboratory model organisms can extend lifespan (Fontana *et al*, 2010; Kenyon, 2010), maintain function in older ages (Barzilai *et al*, 2012) and ameliorate pathology associated with multiple age-related diseases (Niccoli & Partridge, 2012; Johnson *et al*, 2015). Furthermore, lowered activity of the IIS network is implicated in human survival to advanced ages (Suh *et al*, 2008; Tazearslan *et al*, 2011; Deelen *et al*, 2013; He *et al*, 2014). Understanding exactly how reducing IIS network activity ameliorates the effects of ageing could hence pave the way to prevention of ageing-related disease in humans.

The IIS network responds to nutrient availability, growth factors and stress signals to regulate multiple processes, including development, growth, metabolism, stress resistance, reproduction and lifespan (Fontana *et al*, 2010; Kenyon, 2010). These highly pleiotropic phenotypes make it difficult to pinpoint the mechanisms that specifically ameliorate ageing and hence to understand whether they can be separated from other, not necessarily desirable, pleiotropic effects.

Long-lived IIS mutants show a major and tissue-specific rearrangement of RNA transcript expression, as a consequence of alteration of the activity of target transcription factors (TFs; Fontana *et al*, 2010; Kenyon, 2010). In *Caenorhabditis elegans* and *Drosophila*, the single Forkhead boxO, FoxO, transcription factor (DAF-16 and dFOXO, respectively) is required for blunted IIS to extend lifespan (Kenyon *et al*, 1993; Slack *et al*, 2011; Yamamoto & Tatar, 2011), with other TFs also required in both organisms (Kenyon, 2010; Tepper *et al*, 2013; Slack *et al*, 2015). In *Drosophila*, dFOXO is required only for longevity and xenobiotic resistance, and not for other phenotypes of reduced IIS (Slack *et al*, 2011). Hence, in *Drosophila*, *dfoxo* dependence triages at least some of the gene expression

1   Max-Planck Institute for Biology of Ageing, Cologne, Germany
2   CECAD Cologne Excellence Cluster on Cellular Stress Responses in Aging Associated Diseases, Cologne, Germany
3   Department of Proteomics and Signal Transduction, Max-Planck-Institute of Biochemistry, Martinsried, Germany
4   Section of Bioinformatics and Systems Cardiology, Department of Internal Medicine III and Klaus Tschira Institute for Integrative Computational Cardiology, University of Heidelberg, Heidelberg, Germany
5   DZHK (German Centre for Cardiovascular Research), Partner site Heidelberg/Mannheim, Heidelberg, Germany
6   Institute of Healthy Ageing, and GEE, UCL, London, UK
7   Center for Molecular Medicine Cologne (CMMC), University of Cologne, Cologne, Germany
    *Corresponding author. Tel: +49 221 47884021; E-mail: andreas.beyer@uni-koeln.de
    **Corresponding author. Tel: +49 221 37970602; E-mail: partridge@age.mpg.de

changes potentially causal in longevity from several other phenotypes caused by IIS mutants, and facilitates identification of the mechanisms involved.

Profiling of the RNA transcriptome has identified genes and molecular mechanisms that may ameliorate ageing in IIS mutants in *C. elegans* (Murphy *et al*, 2003; Halaschek-Wiener *et al*, 2005; Oh *et al*, 2006; McElwee *et al*, 2007; Ewald *et al*, 2015; Kaletsky *et al*, 2016) and *Drosophila* (Teleman *et al*, 2008; Alic *et al*, 2011). Increased detoxification (Amador-Noguez *et al*, 2007; Selman *et al*, 2009; Ewald *et al*, 2015; Afschar *et al*, 2016) and reduced translation (Selman *et al*, 2009; Afschar *et al*, 2016; Kaletsky *et al*, 2016) are two functional signatures consistently associated with longevity across different model organisms (McElwee *et al*, 2007). However, in general RNA profiling in *C. elegans* has been carried out on whole worms, potentially leaving tissue-specific mechanisms undetected. Furthermore, a wide range of post-transcriptional mechanisms may modify expression of proteins (Barrett *et al*, 2012; Batista & Chang, 2013; Cech & Steitz, 2014; Liu *et al*, 2016; MacInnes, 2016), with recent reports of the correlation between expression levels in the transcriptome and the proteome ranging from 40 to 84% (Wolkow, 2000; Schwanhäusser *et al*, 2011; Li *et al*, 2014).

Tissue-specific modulation of IIS can also extend lifespan, in the worm, *Drosophila* and the mouse (Wolkow, 2000; Blüher *et al*, 2003; Giannakou, 2004; Demontis & Perrimon, 2010; Alic *et al*, 2014). Conditional knockout models in mice, and studies in humans, have revealed that the responses of gene expression to reduced IIS are tissue-specific (Nandi *et al*, 2004; Rask-Madsen & Kahn, 2012). Recently, tissue-specific, global regulation of RNA expression in IIS mutants has been undertaken in the *Drosophila* gut and fat body (Alic *et al*, 2014) and in neurons in *C. elegans* (Alic *et al*, 2014; Kaletsky *et al*, 2016). However, proteomic profiling of individual tissues in the key invertebrate model organisms *Drosophila* and *C. elegans* has been limited, due to their small size, and the requirement for large quantities of starting material. Initial investigations into the proteome of wild-type *Drosophila* focused on protein identification, using samples of low complexity, and either subcellular or biochemical fractionation to achieve increasing depth (Veraksa *et al*, 2005; Brunner *et al*, 2007; Aradska *et al*, 2015). Recent advances in mass spectrometry, sample preparation techniques and data analysis now allow the quantification of near complete proteomes and proteomic expression profiling (Nagaraj *et al*, 2012; Mann *et al*, 2013; Azimifar *et al*, 2014; Kim *et al*, 2014; Deshmukh *et al,* 2015). These new developments have reduced the amount of starting material required for shotgun proteomics and created an opportunity to investigate how attenuated IIS affects the proteome of individual fly tissues and hence to identify candidate mechanisms that ameliorate the effects of ageing.

In this study, we profiled the changes in the tissue-specific proteomes of IIS mutant *Drosophila* and their responses to removal of the key dFOXO transcription factor. We profiled four key insulin-sensitive tissues: the brain, gut, fat body and muscle. The majority of the responses to a systemic reduction in IIS were highly tissue-specific, and 60% of them were not detected in previous transcriptional studies. Proteins associated with the ribosome and the mitochondrial electron transport chain were differentially expressed in the fat body, and led to a reduction in translation and increased mitochondrial respiration. The gut showed a proteomic signature of proteasome–ubiquitin-mediated catabolism, and was associated

with elevated proteasomal assembly and activity. Increased respiration and proteasome activity in the fat body and gut, respectively, were required for the increased longevity of IIS mutants. Importantly, manipulation of mitochondrial biogenesis in the fat body and of proteasomal activity in the gut could each extend lifespan in wild-type flies. Our tissue-specific, proteomic analysis has thus revealed how individual tissues of an organism can act in concert, through diverse tissue-specific responses, to ameliorate the ageing of the whole organism in response to reduced IIS.

# Results

## The tissue-specific proteome of wild-type flies

We first characterized the tissue-specific proteomes of control, wild-type flies. We dissected the brain, thorax (containing predominately muscle), intestinal tract with Malpighian tubules (referred to as gut from here on) and abdominal fat body of young (10 days), adult, female flies. We then performed single-shot, label-free proteomics with a Q-Exactive benchtop quadrupole-Orbitrap mass spectrometer (Fig 1A). We achieved 99% confidence of identification at both protein and peptide levels, and in total identified and quantified 96,404 peptides corresponding to 6,085 proteins, representing 44% coverage of the predicted, protein-coding genes (Fig 1B, Dataset EV1).

The four tissues shared a common core of 1,916 proteins (Fig 1C). Gene ontology (GO) enrichment analysis showed that these served a variety of house-keeping functions, and included mitochondrial and ribosomal constituents (see Dataset EV2 for GO analysis). The four tissues also showed marked differences, with 6–26% of the proteins identified as present in a tissue being unique to it (Fig 1C). To further understand the proteomic features that distinguish individual tissues, we performed principal component (PC) analysis. 74% of the variation between tissues could be captured by the first two PCs. Mapping GO terms onto the proteins contributing to PC1 revealed that proteins with neuronal functions accounted for much of the effect, separating the brain, which had the largest number of unique proteins, from the other three tissues. PC2 detected mainly differences in developmental origins, structure and metabolic activity (Fig 1D and Dataset EV3). Biological replicates from each wild-type tissue clustered tightly together (PCA, Fig 1D), confirming reproducibility (Fig EV1).

## Reduced IIS remodels tissue-specific proteomes

We next identified the effect on the proteomes of down-regulation of IIS. Ablation of the median neurosecretory cells (mNSCs) in the fly brain, which secrete three of the seven *Drosophila* insulin-like peptides (Dilps) into circulation, causes a robust extension of lifespan (Broughton *et al*, 2010). We compared protein expression levels in mNSC-ablated (*InsP3-Gal4/UAS-rpr*) flies to those of wild-type controls (*w^{Dah}*). In total, expression of 2,372 proteins was significantly altered upon reduced IIS (10% FDR, Fig 2 and Dataset EV4), 982 of which showed absolute fold changes larger than two in at least one tissue. 60% of these changes were not previously identified at the RNA transcript level in whole IIS mutant flies (McElwee *et al*, 2007; Alic *et al*, 2011). Thus substantial, new information was gained by tissue-specific proteome analysis.

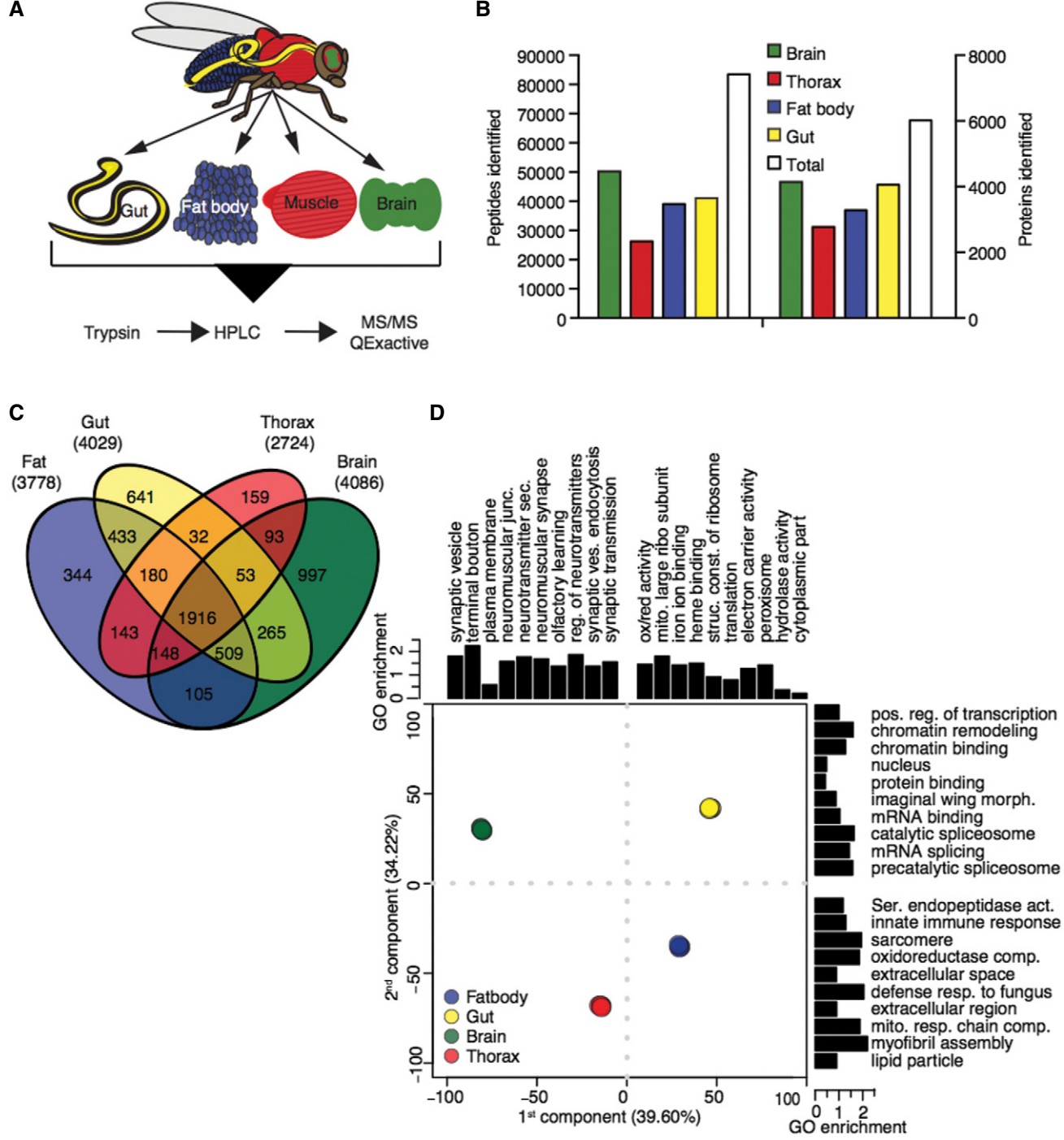

**Figure 1. *Drosophila* tissues contain common and tissue-specific proteomes.**

A   Schematic of experimental design.
B   Tissue-specific and total number of detected peptides and proteins.
C   Venn diagram of proteins detected in wild-type fly tissues.
D   Principal component analysis of the four wild-type tissues GO terms enriched in the 5% tails of contribution along each dimension. Replicates show little variation and overlay each other.

The brain and gut showed the highest numbers of differentially regulated proteins (1,312 and 1,062, respectively), whilst the fat body (256) and thorax (119) showed comparatively few. Changes in expression were almost entirely tissue-specific (85%), with only two proteins, β-tubulin (60D) and l(1)G0230, showing changes in all four (Dataset EV4). Only 13% of the proteins that were

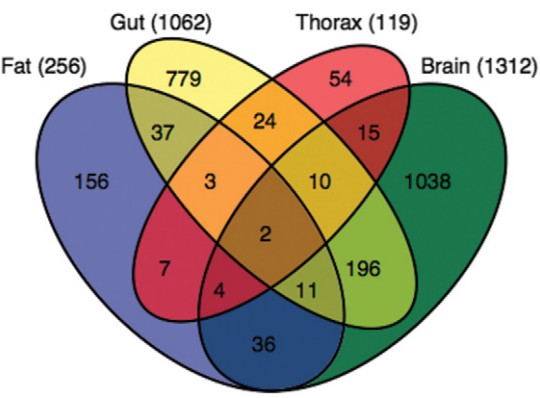

**Figure 2. Proteome response to reduced IIS is highly tissue-specific.**

Venn diagram of differentially expressed proteins upon reduced IIS, comparing tissues from control ($w^{Dah}$) and mNSC-ablated flies. Total number of differentially expressed, tissue-specific proteins is shown in parentheses.

differentially regulated in a single tissue were also detected as solely expressed in that tissue, and thus, the tissue-specific nature of the differential regulation by IIS was mainly due to tissue-specific responses rather than tissue-specific expression patterns (Dataset EV4).

**Identifying tissue-specific, differential protein expression potentially causal in extension of lifespan**

To identify candidate proteins that could be causal in the increased longevity of mNSC-ablated flies, we classified them according to their requirement for *dfoxo* for their differential regulation. We took the set of proteins that changed expression with ablation of mNSCs in wild-type flies and asked whether their response to mNSC ablation was abrogated in a *dfoxo*-null background. Accordingly, we profiled the tissue-specific proteomes of mNSC-ablated flies lacking *dfoxo* (*Insp3-Gal4/UAS-rpr; dfoxo*$^{\Delta/\Delta}$) and the corresponding *dfoxo*$^{\Delta/\Delta}$ controls. We constructed a linear model using all four genotypes and identified proteins whose expression was significantly affected by the interaction between ablation of mNSCs and *dfoxo* deletion. We excluded from this set any proteins whose response to reduced IIS was exaggerated, as opposed to abrogated, in the *dfoxo*-null background. We thus identified 361 proteins that required *dfoxo* for their change in expression in IIS mutant flies (Fig EV2). To assess whether those 361 proteins were more likely to be direct or indirect targets of *dfoxo*, we searched for predicted *dfoxo*-binding motifs within 1 kb of their transcriptional start sites using MEME (Bailey *et al*, 2009). 45% of the 361 proteins came from genes with *dfoxo*-binding motifs (Dataset EV4) and may therefore be directly regulated by dFOXO. However, the remaining 55% of those 361 proteins came from genes lacking *dfoxo*-binding motifs which suggest that although their expression was *dfoxo*-dependent, they were not directly regulated transcriptionally by dFOXO.

The proteins whose expression showed no significant interaction between response to mNSC ablation and the presence of *dfoxo* could have been truly *dfoxo*-independent or false negatives. To define a set of proteins that are most likely *dfoxo*-independent, but ablation responsive, we identified those for which the fold difference in responsiveness to mNSC ablation in the wild-type and *dfoxo*-null background was significantly less than the minimum fold change of *dfoxo*-dependent proteins in the same tissue. Additionally, we included the proteins that showed an exaggerated responsiveness to the ablation in the *dfoxo*-null background. In this way, we defined a set of 196 proteins as *dfoxo*-independent, with high confidence. Indeed, 52% of these showed significant differences in expression even between the *dfoxo*$^{\Delta/\Delta}$ and *dfoxo*$^{\Delta/\Delta}$ *Insp3-Gal4/UAS-rpr* (Fig EV2).

To further increase the power of our analysis of *dfoxo* dependency, we employed network propagation, which combines differential expression with physical interactors of individual proteins to identify altered subnetworks in each tissue (Vanunu *et al*, 2010). Differentially expressed *dfoxo*-dependent or *dfoxo*-independent proteins were mapped independently onto the *Drosophila* protein–protein interaction network (Murali *et al*, 2011), and the associated *P*-value of each protein was negative logarithmic transformed and propagated to adjacent interacting proteins. We then clustered the *dfoxo*-dependent and *dfoxo*-independent responses to reduced IIS in each tissue and identified functional categories of these clusters with GO enrichment analysis (Dataset EV5).

We focused on clusters whose regulation was predicted as *dfoxo*-dependent within the network propagation analysis, since these should include processes causally linked to longevity. *dfoxo*-dependent responses in the brain were enriched for mitochondrial electron transport chain, mRNA splicing and nucleosome components (Fig 3), whilst in the gut they were enriched for proteasome and ubiquitin-mediated protein catabolism (Fig 3). Mitochondrial electron transport chain was also enriched within *dfoxo*-dependent responses in the fat body, as were ribosomal constituents and proteins involved in nucleosome assembly. Amongst the functional groups identified by network propagation, we further focused on three strong candidates for mediating the longevity of IIS mutant flies. First, we chose the ribosome, which we identified as an almost uniquely fat body-specific response. Second, we focused on the mitochondrial electron transport chain, the only response identified as common to multiple tissues, the brain, gut and fat body. Finally, we examined proteasome–ubiquitin protein catabolism as the most enriched cluster in the gut.

**Reduced IIS alters fat body-specific, *dfoxo*-dependent protein expression to regulate translation**

We next determined if the tissue-specific, *dfoxo*-dependent proteomic responses led to tissue-specific functional changes. Our proteomic and bioinformatics analysis identified a fat body-specific, *dfoxo*-dependent reduction in ribosome-associated proteins in response to reduced IIS (Fig 3). To determine whether this signature was reflected in altered physiology, we quantified tissue-specific translation, using $^{35}$S incorporation to detect *de novo* protein synthesis. Individual tissues were dissected and incubated in medium containing $^{35}$S-labelled methionine and cysteine. We separated samples by SDS–PAGE and quantified *de novo* labelled proteins (Fig EV3). No significant change was detected in head, thorax or gut (Fig EV3), but there was significantly less incorporation of labelled amino acids into proteins in the fat body of the IIS mutants,

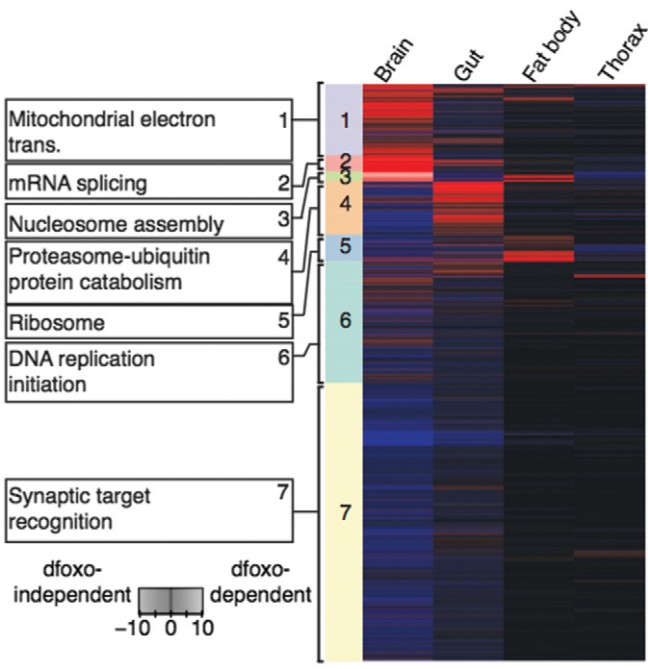

**Figure 3.  Hierarchical clustering and GO enrichment analysis of dFOXO-dependent and dFOXO-independent IIS-mediated regulation of the proteome.**

Tissue-specific heatmap of significantly regulated *dfoxo*-dependent (red) and *dfoxo*-independent (blue) proteins in response to reduced IIS. Coloured side bars represent network propagated score clustering and associated most significantly enriched GO terms.

and that reduction was dependent on *dfoxo* (two-way ANOVA $P < 0.05$; Fig EV3). This finding suggests that IIS-mediated, *dfoxo*-dependent reduction in translation in the fat body may contribute to longevity.

## Reduced IIS tissue-specifically and *dfoxo*-dependently regulates respiration

Changes in mitochondrial function have been linked to ageing in several model organisms (Bratic & Larsson, 2013). Paradoxically, however, both reduced mitochondrial respiratory chain protein levels (Copeland *et al*, 2009) and increased mitochondrial biogenesis (Rera *et al*, 2011) can extend lifespan. Our analysis highlighted a tissue-specific regulation of mitochondrial proteins that was *dfoxo*-dependent in the brain and fat body and independent of *dfoxo* in the gut (Fig 3). In total, we detected 60 proteins belonging to the annotation mitochondrial electron transport chain, with 6–35% of them in different tissues showing significantly altered levels upon IIS reduction (Fig 4A). They were coordinately up-regulated in the fat body, with no clear coordinated change in other organs (Fig 4A). Using Western blotting, we directly confirmed that NDUFS3, a complex I subunit, was up-regulated in a *dfoxo*-dependent manner in the fat body (Fig 4B).

To determine the functional significance of the observed changes in protein levels, we provided substrates for the individual complexes of the electron transport chain to measure respiration in actual respiratory state (PGMP3), the potential maximum

respiratory state (Vmax) and role of complex I (rotenone sensitive). In agreement with our proteomic and bioinformatic analysis, endogenous respiration (PGMP3) was up-regulated by reduced IIS in the fat body, a response that was completely dependent on *dfoxo* (two-way ANOVA interaction term $P < 0.01$; Fig 4C). This was also the case for the Vmax and rotenone-sensitive respiratory states, suggesting that the increased respiration was due to increased capacity of the respiratory chain, at least in part through complex I (Fig 4C). In contrast, endogenous (PGMP3) and Vmax respiratory states in the gut were reduced in mNSC-ablated flies and, as predicted by our analysis of the proteome, this occurred independently of *dfoxo* (Fig 4C). Respiration in the head and thorax was unchanged by reduced IIS (Fig 4C). We confirmed the increase in endogenous respiration of the fat body, and the reduced endogenous respiration of the gut, in an independent model of reduced IIS, long-lived *dilp2-3,5* mutant flies (Gronke *et al*, 2010; Fig 4D).

To further understand the mechanistic basis for increased respiration in the fat bodies of mNSC-ablated flies, we examined mitochondrial biogenesis. We first examined mitochondrial DNA (mtDNA) levels in the fat body, in the presence and absence of *dfoxo*, and found that they were increased in the MNC-ablated flies in a *dfoxo*-dependent manner (two-way ANOVA interaction term $P = < 0.01$; Figs 4E and EV4A). Mitochondrial biogenesis is driven by nuclear transcription factors, including nuclear respiratory factors 1 and 2 (nrf; Evans & Scarpulla, 1990; Virbasius *et al*, 1993), which are co-activated by PGC-1 to regulate the expression of mitochondrial proteins (Puigserver & Spiegelman, 2003; Scarpulla, 2008; Tiefenböck *et al*, 2010). *Drosophila* has single homologs of PGC-1 and nrf-2 (Gershman *et al*, 2007), named *spargel* and *delg*, respectively (Tiefenböck *et al*, 2010), and loss of either reduces transcription of nuclear encoded mitochondrial genes (Tiefenböck *et al*, 2010). *Spargel* and *delg* are both expressed at low levels in adult *Drosophila* fat body (Graveley *et al*, 2011), and we did not detect them, or their regulation in our analysis (Dataset EV1). However, as with many TFs and cofactors, the activity of Spargel and delg could be regulated on the post-translational level (Li *et al*, 2007). We tested whether *spargel* or *delg* were required for the increased respiration in the fat bodies of IIS mutant flies. To achieve fat body-specific knockdown of *spargel* and *delg*, we performed this analysis in *dilp2-3,5* mutants. We found that increased respiration in the fat body of the IIS mutant was entirely dependent on *Spargel* and *delg* expression in the fat body (two-way ANOVA interaction term $P = < 0.001$; Fig EV4B and C). Furthermore, knockdown of expression of *Spargel* or *delg* in the fat body of *dilp2-3,5* mutants reduced the extent of their increased longevity, but did not affect the lifespan of wild-type controls (Figs 4F, and EV4D and E). Experimentally increased expression of *Spargel* in the adult fat body, using the GeneSwitch driver S106-GS, was sufficient to increase respiration (Fig EV4F) and lifespan (Figs 4G and EV4G), whilst over-expression in the gut alone was not (Fig EV4H).

Overall, our data reveal that lowered systemic IIS coopts different mechanisms to regulate respiration in different tissues. Reduced IIS decreased respiration in the gut independently of *dfoxo*, whilst simultaneously increasing respiration in the fat body in a *dfoxo*- and *spargel/delg*-dependent manner. Furthermore, increased mitochondrial biogenesis, and thus respiration, in the fat body is both necessary and sufficient to extend lifespan.

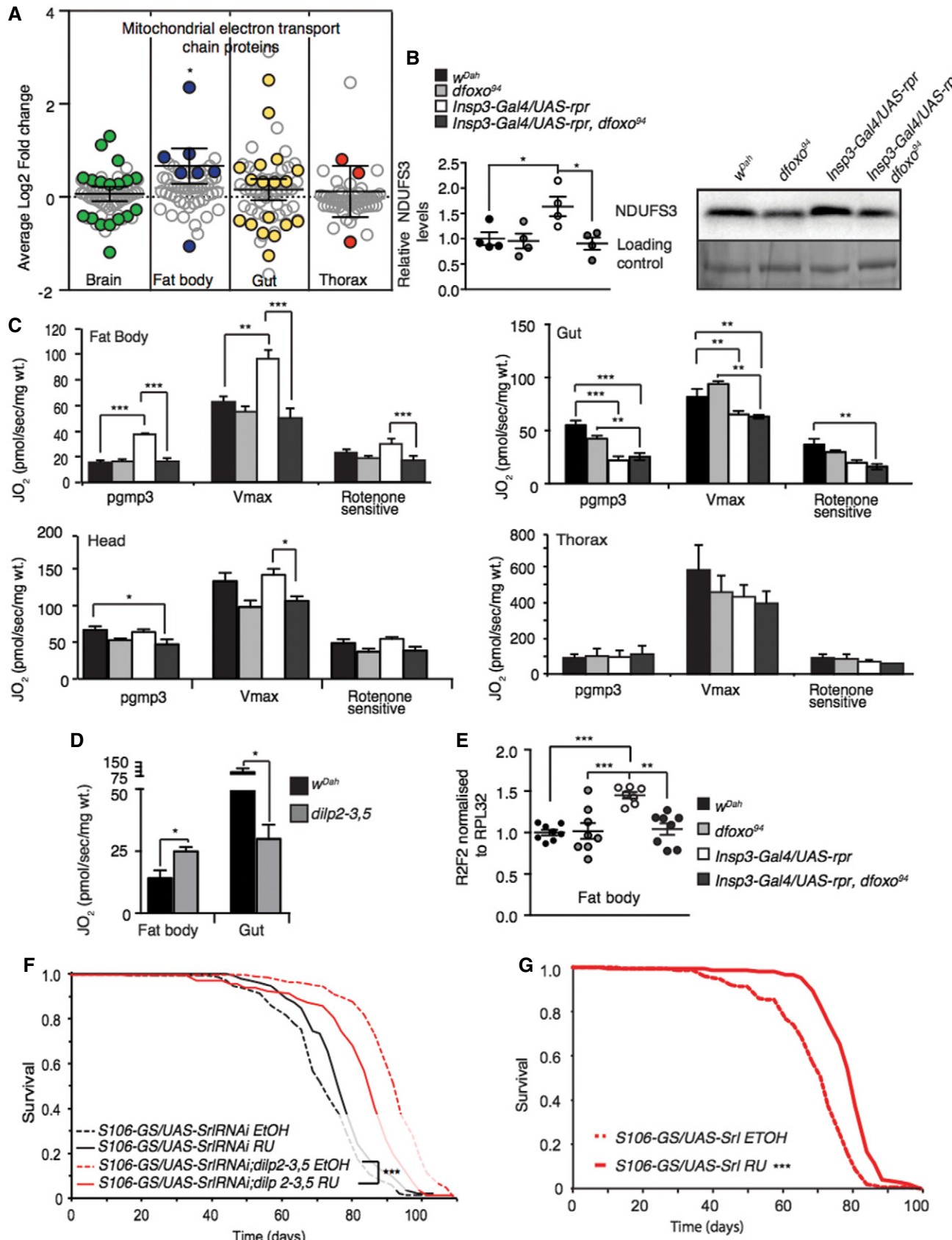

**Figure 4.**

◀

**Figure 4.  Reduced IIS differentially regulates tissue-specific mitochondrial function and biogenesis.**

A    Average log2 fold change of IIS-regulated proteins associated with the functional term mitochondrial electron transport chain. One-sample *t*-test shows directional significance. Grey circles show all detected proteins, and coloured circles show significantly regulated proteins.

B    Western blot analysis of fat body NDUFS3, and stain free loading control (*n* = 4), significance determined by *t*-test.

C    Fat body, gut, head and thorax oxygen consumption normalized to dry wt. in 10 days old control, *dfoxo*[Δ94] mutants, mNSC ablation, mNSC-ablated flies lacking *dfoxo*. Oxygen consumption was assessed by using substrates entering the level of complex I (PGMP3), complex I+II once uncoupled by CCP (Vmax) and rotenone-sensitive complex I+II + rotenone (*n* = 5).

D    Respiratory measurements of an independent IIS mutant (*dilp2-3,5*) compared to controls (*w*[Dah]) (*n* = 5), significance determined by *t*-test.

E    Relative mtDNA levels compared to nuclear DNA in fat body (*n* = 8). Relative mtDNA levels in the gut, head and thorax are shown in Fig EV4A. Significance determined by one-way ANOVA.

F    Lifespan analysis of *dilp2-3,5* mutant and control flies with reduced expression of *Spargel* in the fat body (*S106-GS/UAS-SrlRNAi*) induced by RU or none induced (*n* = 150/genotype/treatment). Lifespan analysis of genetic controls shown in Fig EV4D. Statistical significance was determined by Log Rank test.

G    Lifespan analysis of flies over-expressing *Spargel*, in the adult fat body (*n* = 150/genotype/treatment). Genetic control lifespan shown in Fig EV4G.  Statistical significance was determined by Log Rank test.

Data information: Bars indicate mean ± SEM (*$P < 0.05$; **$P < 0.01$; ***$P < 0.001$; *n* = 5).

## Reduced IIS alters the *dfoxo*-dependent gut proteome, increasing proteostasis to maintain gut health

The proteasome is a major site of protein degradation in the cell (Lecker *et al*, 2006), and its dysfunction is associated with ageing and disease (Vilchez *et al*, 2014). Our proteomic and bioinformatics analysis identified a prominent, gut-specific, *dfoxo*-dependent regulation of proteasomal proteins in response to reduced IIS (Fig 3). The 26S proteasome consists of 33 proteins, 14 of which (7-α & 7-β subunits) constitute the 20S proteasome core, whilst 19 constitute the 19S regulatory particle (Tomko & Hochstrasser, 2013). In total, we detected 94% of the proteasomal subunits, and 36% were differentially regulated specifically in the gut in response to reduced IIS (Fig 5A). However, we did not detect a consistent pattern or direction of regulation (4 up- and 8 down-regulated, Fig 5A). Proteasomal subunit Rpt6R showed the greatest degree of regulation, increasing 2.6-fold in the gut. We confirmed this change and its *dfoxo* dependency, by Western blot analysis of the guts of mNSC-ablated flies compared to controls (Fig 5B). Rpt6, a 19S proteasomal subunit, plays a key role, along with Rpn6, in regulating the assembly and activity of the 26S proteasome holoenzyme (Park *et al*, 2011; Pathare *et al*, 2012; Sokolova *et al*, 2015). Hence, up-regulation of this subunit could indicate an up-regulation of proteasomal assembly. Indeed, the guts of mNSC-ablated flies showed a threefold increase in assembly of the 26S proteasome compared to controls (Fig 5C), as measured using an in-gel assay (Vernace *et al*, 2007). These data suggest that reduced IIS results in increased proteasomal assembly in the gut, possibly through increased levels of Rpt6.

We next determined whether the increased proteasomal assembly in the gut affected proteasomal function. To measure proteasome activity, isolated tissue homogenates were incubated with a fluorogenic (AMC, 7-Amino-4-methylcoumarin) proteasome substrate (Z-Leu-Leu-Glu-AMC), allowing the quantification of peptidylglutamyl-peptide hydrolysing (caspase-like) activity of the proteasome. As predicted by our bioinformatic analysis (Fig 3), no differences in activity were detected in the fat body, head or thorax in response to reduced IIS (Fig 5D). However, in agreement with our assembly data, reduced IIS resulted in a highly significant, *dfoxo*-dependent, threefold increase in proteasomal activity in the gut (two-way ANOVA interaction term $P < 0.001$; Fig 5D). To establish the generality of this finding, we measured proteasomal activity in long-lived *dilp2-3,5* mutants (Gronke *et al*, 2010) and found

that, again, proteasome activity was increased in gut but not fat body (Fig 5E). Thus, increased proteasome activity in the gut in response to reduced IIS is a candidate mechanism for increased longevity.

To determine the functional relevance of gut-specific increased proteasomal activity, we quantified the level of ubiquitinated proteins. Ubiquitination serves many functions in the cell (Grabbe *et al*, 2011). Poly-ubiquitination, specifically on Lysine 48 (K48), however, targets the protein for degradation by the proteasome, and accumulation of poly-ubiquitinated proteins, often seen with increasing age, indicates loss of proteostasis (Tonoki *et al*, 2009). Based on our tissue-specific proteasomal activity assay (Fig 5D), we hypothesized that K48-ubiquitinated proteins levels would not differ in the head, thorax or fat body as a result of altered IIS activity, but would be reduced in the gut. We performed Western blot analysis against K48 poly-ubiquitinated proteins (Fig 5F) and, in agreement with our hypothesis, we detected a gut-specific, *dfoxo*-dependent reduction in K48 poly-ubiquitinated proteins in response to reduced IIS (Fig 5F). Therefore, increased proteasomal assembly/activity, and thus clearance of K48 poly-ubiquitinated proteins, may underlie IIS-mediated longevity through enhanced proteome maintenance.

## Increased gut proteasomal activity maintains gut integrity and extends lifespan

We next determined the role of increased proteasomal activity in the extended lifespan of IIS mutants. We orally administered bortezomib, a potent inhibitor of the proteasome, to flies with reduced IIS and controls, for the duration of their lifespan. At a concentration of 2 μM, bortezomib did not reduce the lifespan of wild-type flies (Fig 6A), but significantly reduced the lifespan-extension of mNSC-ablated flies (Fig 6A) and, independently, of *dilp2-3,5* mutants (Fig EV5A). Using Cox proportional hazards, we confirmed that treatment with bortezomib had a significantly different effect on survival in the two models of reduced IIS flies compared to their wild-type controls ($P < 0.01$). Higher concentrations of bortezomib (3 μM) completely abolished the difference in lifespan between wild-type and ablated flies, but also reduced the survival of wild-type flies (Fig EV5B).

To determine the functional consequence of increased proteasome activity in the gut, we quantified age-associated changes in gut integrity in control and long-lived flies. Feeding non-absorbable

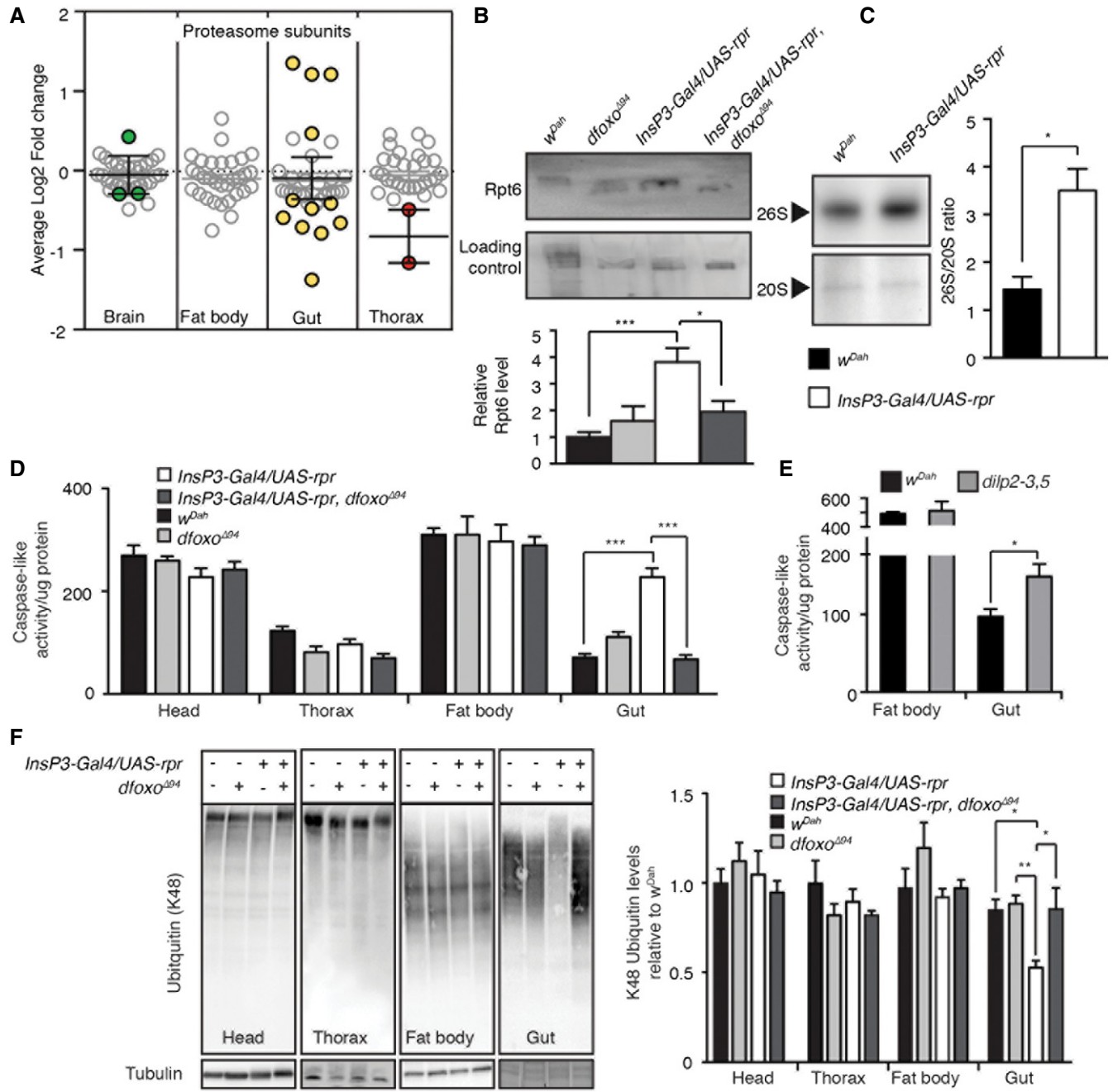

**Figure 5.  IIS-*dfoxo*-dependent regulation of proteasome activity in the gut enhances proteome maintenance.**

A   Tissue-specific average log2 fold change of IIS-regulated proteins associated to the proteasome.

B   Western blot and quantification analysis of Rpt6 in wild-type (*w^Dah*) (n = 8), *dfoxo^Δ94* mutants (n = 3), mNSC-ablated flies (*InsP3-Gal4/UAS-rpr*) (n = 7) and mNSC-ablated flies lacking *dfoxo* (*InsP3-Gal4/UAS-rpr, dfoxo^Δ94*) (n = 4), normalized to protein loading and significance determined by one-way ANOVA.

C   Assessment of proteasome assembly by in-gel caspase-like (LLE) proteasome activity in mNSC-ablated fly gut and control flies (*w^Dah*). Quantification shows the activity ratio 26S/20S (n = 5), significance determined by *t*-test.

D   Tissue-specific caspase-like activity was assessed using fluorogenic substrate (LLE-AMC, Enzo Life Science) in wild-type (*w^Dah*; n = head 14, fat body 5, gut 8, thorax 8), *dfoxo^Δ94* mutants (n = head 9, fat body 5, gut 7, thorax 8), mNSC-ablated flies (*InsP3-Gal4/UAS-rpr*; n = head 14, fat body 5, gut 7, thorax 9) and mNSC-ablated flies lacking *dfoxo* (*InsP3-Gal4/UAS-rpr, dfoxo^Δ94*; n = head 7, fat body 5, gut 9, thorax 9), significance determined by two-way ANOVA and *post hoc* pairwise tests.

E   Proteasomal caspase-like activity of an independent IIS mutant (*dilp2-3,5,* grey) compared to controls (*w^Dah*) (gut n = 9, fat body n = 7). Statistical significance was determined by *t*-test.

F   Representative Western blot showing tissue-specific levels of K48-linked poly-ubiquitin and their quantification (head, thorax and fat body Westerns performed in quadruplicate). For gut samples wild-type (*w^Dah*) and *dfoxo^Δ94* mutants n = 11, *InsP3-Gal4/UAS-rpr* n = 12 and *InsP3-Gal4/UAS-rpr, dfoxo^Δ94* n = 10, significance determined by two-way ANOVA and *post hoc* pairwise tests.

Data information: *P < 0.05, **P < 0.01, ***P < 0.001. All error bars show SEM.

blue food dye to flies allows direct assessment of gut integrity, with loss of integrity resulting in blue dye leaking into the hemolymph giving "smurf" flies (Rera *et al*, 2012). As previously reported (Rera *et al*, 2012), control flies show an age-associated loss of gut integrity, with the proportion of "smurfs" in the population increasing with age; however, this was suppressed in mNSC-ablated flies (Fig 6B), and independently in *dilp2-3,5* mutants (Fig EV5C). Whilst treatment with bortezomib (2 μM) did not affect aged control flies, it dramatically increased the proportion of "smurfs" in mNSC-ablated flies (Fig 6C). These effects of the proteasomal inhibitor on lifespan and gut integrity indicate that increased proteasomal activity in the gut is required for the lifespan extension of IIS mutants because it increases gut integrity.

Together, the *dfoxo* dependence of the increased proteasomal activity and increased gut integrity of mNSC-ablated flies, as well as the known role of proteostasis in ageing (Lopez-Otin *et al*, 2013), prompted us to experimentally test whether increasing proteasomal assembly and activity in the adult gut was sufficient to extend lifespan. RPN6 aids in stabilizing the assembly of the 26S proteasome (Pathare *et al*, 2012) and can increase lifespan when over-expressed ubiquitously in *C. elegans* (Vilchez *et al*, 2012). We therefore over-expressed RPN6 specifically in the gut, using the constitutive, midgut-specific Gal4 driver *Np1*, which drives expression exclusively in enterocytes (Jiang *et al*, 2009; Alic *et al*, 2014). Constitutive, gut-specific over-expression of RPN6 was sufficient to significantly increase proteasome assembly in the gut (Fig 6D), increase proteasomal activity (Fig 6E) and induce a 26% reduction in K48 poly-ubiquitinated proteins (Fig 6F), suggesting increased protein clearance and increased gut integrity (Fig 6G). Thus, increased levels of RPN6 are sufficient to increase proteasome assembly and activity, and increased clearance of K48 poly-ubiquitinated proteins, enhancing proteome maintenance, and thus the health of the gut.

We next determined whether over-expression of RPN6 was sufficient to extend the fly lifespan. Constitutive, gut-specific over-expression of RPN6 resulted in a significant 5% increase in mean lifespan, and up to a 19% increase in maximal lifespan compared to controls (Fig 6H). We independently confirmed this finding by over-expressing RPN6 in the adult gut using the inducible, gut-specific GeneSwitch driver *TIGS-2*, which was sufficient to increase proteasome activity, increase gut integrity and induce a small, but significant, extension of lifespan (Figs 6I and EV6A–C). Hence, increased gut proteome maintenance, through increased proteasome function, can increase gut integrity and extend lifespan.

# Discussion

Reduced activity of the evolutionarily conserved insulin/IGF-like signalling network extends longevity and reduces age-associated pathologies, extending health and vitality (Fontana *et al*, 2010). Several studies have examined transcriptome remodelling in response to reduced IIS, to uncover the genes and molecular mechanisms underlying the longevity phenotype. However, the specific combination of expression changes and physiological processes that are required to prolong healthy lifespan remains elusive. Here, we have quantified the tissue-specific proteome of *Drosophila* in response to reduced IIS, using genetic manipulations that allowed us to focus on expression profiles associated with longevity.

## Establishing the tissue-specific proteome of *Drosophila*

In recent years, the technology for proteomics has begun to reach the accessibility, sensitivity and reproducibility of genomic studies (Mann *et al*, 2013). Taking advantage of advancing technologies, we have performed an in-depth, tissue-specific proteomic screen in *Drosophila*, detecting over 6,000 proteins, ~40% of the predicted proteome. This dataset emphasizes the importance of tissue-specific profiling and provides a unique and freely available resource for future expression studies in *Drosophila* (Dataset EV1).

Our proteomic analysis revealed that ~15% of the predicted fly proteome is remodelled in response to reduced IIS, with 2,372 proteins showing significant up- or down-regulation. The RNA transcripts encoding 60% of these proteins have not previously been identified as differentially regulated by reduced IIS in whole fly profiles. This may be due to the highly tissue-specific nature of the changes, with only two proteins being regulated in all tissues. Differences between patterns of expression of RNA and protein may also be important, due to post-transcriptional regulation and changes in protein stability (Liu *et al*, 2016). For example, expression of GCN4 in nutritionally challenged yeast increases due to increased translation, not transcription (Ingolia *et al*, 2009), whilst in flies subject to dietary restriction several mitochondrial genes are regulated by translational, not transcriptional, control (Zid *et al*, 2009).

Confirming the quality of this dataset, proteins associated with several IIS-mediated processes, including development, growth, sleep, lipid metabolism, translation and adult lifespan, were detected and differentially regulated. We were able to pinpoint these responses to specific tissues. For example, global reduction in translation can extend lifespan in *C. elegans* and *Drosophila* (Hansen *et al*, 2007; Pan *et al*, 2007; Wang *et al*, 2014) and is an evolutionarily conserved response to reduced IIS (McElwee *et al*, 2007; Stout *et al*, 2013; Essers *et al*, 2016), and our study showed this response to be specific to the fat body.

Our dataset also identified possible novel mediators of responses to reduced IIS. For example, our proteomic analysis suggested a gut-specific regulation of proteasomal function in response to reduced IIS, which was confirmed by finding a corresponding gut-specific proteasomal phenotype. We have also characterized *dfoxo*-dependent changes in protein expression in IIS mutant flies, separating those changes associated with IIS-mediated longevity from those changes associated to other IIS-mediated phenotypes. Furthermore, we determined which IIS responsive protein-coding genes contain predicted dFOXO-binding motifs, identifying possible direct and indirect targets of dFOXO. Additionally, we examined transcriptional changes in several regulated candidate genes associated with mitochondria and the proteasome (Fig EV7A–C). Some candidate genes were regulated in a *dfoxo*-dependent manner, consistent with a direct regulation by dFOXO; however, many did not reflect the changes seen at the protein level, suggesting indirect regulation by dFOXO, possibly though post-transcriptional and/or post-translational regulation (Fig EV7A–C). Importantly, we have thus identified longevity-associated changes in the proteome of IIS mutant flies and the processes that they regulate, and we have experimentally demonstrated the role of some of these processes in extension of lifespan.

Figure 6.

**Figure 6.  Proteasome activity is necessary for IIS-mediated longevity, and sufficient to extend lifespan.**

A       Lifespan analysis of wild-type flies and mNSC-ablated flies treated with 2 µM proteasome inhibitor (bortezomib) or vehicle (EtOH) (n = 100).
B, C    Assessment of age-related gut integrity through "Smurf" assay at either different ages (10, 30, 65 days, n = 100), significance determined by t-test, or in aged (55 days, n = 90) flies in the presence of 2 µM proteasome inhibitor (bortezomib), two-way ANOVA.
D       Assessment of proteasome assembly by in-gel caspase-like (LLE) proteasome activity in NP1-Gal4/UAS-RPN6 (n = 3) flies compared to +/UAS-RPN6 control flies (n = 4), significance determined by t-test. Quantification shows the activity ratio 26S/20S (n > 4).
E       Gut proteasome activity (caspase-like) of NP1-Gal4/UAS-RPN6 (n = 4) flies compared to +/UAS-RPN6 control flies (n = 3), significance determined by t-test.
F       Representative Western blot showing gut-specific levels of K48-linked poly-ubiquitin and their quantification (n = 6), significance determined by one-way ANOVA.
G       Assessment of age-related gut integrity through "Smurf" assay at difference ages (10, 30, 65 days) in NP1-Gal4/UAS-RPN6 flies and +/UAS-RPN6 control flies (n > 100), significance determined by t-test.
H, I    Lifespan analysis of flies with gut-specific over-expression of UAS-RPN6 using a constitutive gut-specific driver (Np1-Gal4) or in the adult using Tigs-Gal4-GS in the presence of RU486 (200 µm) or vehicle (EtOH) (n = 150).

Data information: *P < 0.05, **P < 0.01, ***P < 0.001. All error bars show SEM. Statistical significance between lifespan analyses (A, H, I) was determined by Log Rank test.

## Tissue-specific regulation of mitochondrial number and respiration as a regulator of the IIS longevity response

Mitochondrial respiration and the production of reactive oxygen species (ROS) have been proposed as a mechanism of ageing (Bratic & Larsson, 2013). According to this theory, ROS are produced as a by-product of mitochondrial respiration and on interaction with cellular macromolecules they induce damage (Harman, 1956). However, the link between mitochondria and ageing is not simple, and the validity of this theory has been questioned (Bratic & Larsson, 2013).

Seemingly in agreement with reduced ROS being beneficial, reducing mitochondrial respiration extends lifespan in several organisms (Dillin et al, 2002; Rea et al, 2007; Copeland et al, 2009). In contrast, many pro-longevity interventions are associated with increased activation of mitochondrial respiration (Evans & Scarpulla, 1990; Virbasius et al, 1993; Ristow & Schmeisser, 2011), including models of dietary/caloric restriction (Mootha et al, 2003; Baker et al, 2006) and IIS, as found here and in *C. elegans* (Zarse et al, 2012). Declining mitochondrial number and function is associated with ageing (Yen et al, 1989; Shigenaga et al, 1994; Hebert et al, 2015). Thus, increasing or maintaining mitochondrial biogenesis with age may be beneficial.

We found increased, fat body-specific respiration that was causal in the longevity of IIS mutants. Furthermore, we and others (Rera et al, 2011), have shown that increasing mitochondrial biogenesis, extends lifespan, although we found the effect to be confined to the fat body, whilst Rera et al (2011) found an effect in both the fat body and the gut, the discrepancy possibly due to the use of different fly stains. We also found that the increased fat body-specific respiration and longevity of IIS mutant flies are dependent on PGC1-α in the fat body, indicating that PGC1-α is a downstream mediator of IIS, not only for growth (Tiefenböck et al, 2010; Mukherjee & Duttaroy, 2013), but also, as found here, longevity.

How increased respiration can, directly or indirectly, extend lifespan remains unclear. Growing evidence suggests that ROS, produced by mitochondrial respiration, is increased by pro-longevity interventions and may act as signalling molecules to activate the expression of pro-longevity genes (Ristow & Schmeisser, 2011). On the other hand, inhibition of target of rapamycin (TOR) in yeast increases respiration, but reduces ROS production, due to increased electron transport chain subunit expression and increased efficiency of electron transport, preventing a protracted electron transit time and ROS formation (Bonawitz et al, 2007). Furthermore, in flies,

increased mitochondrial biogenesis protects against high fat diets, preventing lipid accumulation and heart dysfunction (Diop et al, 2015). To determine the exact mechanisms involved, analysis of tissue-specific determination of ROS production and electron transport chain complex status under impaired IIS and other pro-longevity interventions will be needed.

Interestingly, although not associated with longevity, we also found that respiration in the gut is decreased under reduced IIS, independently of *dfoxo* activity. Our analysis suggested that this response was associated with down-regulation of Complex III protein subunits. When Complex III subunits are ubiquitously reduced throughout development and adulthood, flies become considerably longer lived and produce fewer offspring (Copeland et al, 2009). However, adult-specific expression fails to extend lifespan, yet flies remain less fecund (Copeland et al, 2009). Since the down-regulation of respiration we observed on reducing IIS in the gut was not associated with increased lifespan, and occurred independently of *dfoxo*, the reduction in gut respiration may underlie the reduced fecundity of IIS mutants.

## Tissue-specific proteostasis as a regulator of the IIS longevity response

Gradual loss of proteostasis is considered a hallmark of ageing (Lopez-Otin et al, 2013). We found that, under reduced IIS, *Drosophila* tissue specifically regulate proteostasis, an evolutionarily conserved response that correlates with longevity. In flies, the age-associated decline of proteostasis has been linked to declining proteasome function (Tsakiri et al, 2013), whilst maintaining proteasome function is associated with longevity (Chondrogianni et al, 2000; Pérez et al, 2009; Vilchez et al, 2012; Ungvari et al, 2013). In yeast, *C. elegans*, and *Drosophila*, ubiquitous over-expression of proteasomal subunits can extend lifespan, possibly through increased maintenance of the cellular proteome (Chen et al, 2006; Tonoki et al, 2009; Kruegel et al, 2011; Vilchez et al, 2012). Increasing the maintenance of the cellular proteome and extending life span can also be achieved by reducing translation (Hansen et al, 2007; Pan et al, 2007; Wang et al, 2014). We found in *Drosophila* that increased proteasome activity and decreased translation correlated with longevity. However, these responses were highly tissue-specific. Only the fat body of *Drosophila* responded to reduced IIS by decreasing translation, and only the gut responded by increasing proteasome activity. We were able to recapitulate increased proteasome activity by gut-specific over-expression of

RPN6, which was sufficient to extend lifespan in wild-type flies. Importantly, we could also show that increased proteasomal activity is required for the longevity of IIS mutants. Understanding how manipulation of one, or relatively few, protein subunits affects the function of a large protein complex is important for translating functional studies towards amelioration of the effects of human ageing. Whilst each subunit plays a role, only some subunits, that is the catalytic subunits, are essential for proteasome activity. However, Rpt3, Rpt6 and RPN6 play an equally vital role, those of opening the catalytic core and assembly maintaining structural integrity of the proteasome (Murata *et al*, 2009; Tian *et al*, 2011; Pathare *et al*, 2012; Sokolova *et al*, 2015). RPN6 over-expression also influences lifespan (Vilchez *et al*, 2012), and we suggest through increased 26S proteasomal assembly, which recapitulates the phenotype seen in IIS mutant flies. It will therefore be important to determine whether similar regulation occurs in the tissues of long-lived mammalian IIS models.

## Tissue-tissue communication and rebuilding the fly

Tissue-specific reduction in IIS activity can improve function in that specific tissue, and can also act at a distance to improve the function of other tissues through systemic effects (Zhang *et al*, 2013; García-Cáceres *et al*, 2016; Kaletsky *et al*, 2016). In these cases, tissue-to-tissue communication, involving FoxO and other key regulators, promotes longevity (Murphy *et al*, 2007; Bai *et al*, 2012; Zhang *et al*, 2013; Alic *et al*, 2014). The fly genome encodes ~300 proteins predicted to be targeted for secretion. We found 35% of these proteins to be regulated in response to reduced IIS, but very few dependent upon *dfoxo*, and thus associated with longevity. Future investigations to directly assay the proteomic changes in circulating hemolymph will help to determine which proteins may be involved in communication between tissues.

A system-level approach to analyse the tissue-specific proteome response to reduced IIS, followed by functional genetic and molecular analysis, has allowed the identification of IIS-mediated, tissue-specific pro-longevity processes. Despite their highly specific responses to lowered IIS, individual tissues each contributed to the increase in lifespan in the IIS mutant flies. As well as its effect on ageing, IIS co-ordinates tissue-specific processes to regulate early life traits such as development, growth and reproduction in response to nutrition and other cues. The level of IIS appears to have evolved to optimize these early life fitness traits, but evidently this level is too high for optimal health at later ages not normally subject to natural selection, resulting in pro-longevity effects of reduced IIS.

Here, we focused on two pro-longevity effects, gut-specific proteasome assembly/activity and fat body-specific respiration/mitochondrial biogenesis. Increasing proteasome activity in the gut was sufficient to extend lifespan by 11%, and elevating mitochondrial respiration in the fat body can extend lifespan to the same extent (11%). However, reducing IIS extends lifespan to a greater extent. For example, mNSCs-ablated flies are ~30% longer lived than controls. Thus, it is tempting to suggest that increasing both proteasome activity and mitochondrial respiration in the same fly could lead to an additive increase in lifespan, and that manipulation of other processes identified here, such as reduced translation in the fat body, may extend lifespan even further (Fig 7), a hypothesis that should be tested in future experiments.

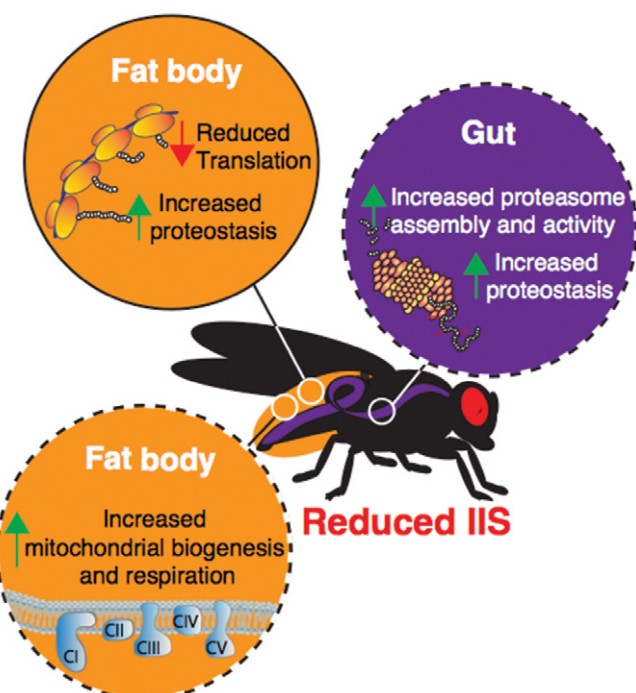

**Figure 7. Model of tissue-specific responses to reduced IIS that may mediate longevity.**

Colours denote different tissues (fat body, orange; gut, purple). Dashed lines show tissue-specific responses that are both necessary and sufficient to extend lifespan.

In summary, tissue-specific proteomic analysis of the responses to reduced IIS in *Drosophila* revealed that proteins from 15% of *Drosophila's* protein-coding genes were regulated in response to reduced IIS across the four tissues examined here. Surprisingly, our study detected considerably more regulated proteins than our previous RNA transcript analyses in IIS mutants, highlighting the importance of tissue-specific, and proteome, profiling to understand the effects of IIS. Here we focused specifically on those underlying responses to reduced IIS that are causal for longevity. Reducing translation, specifically in the fat body, is associated with longevity and not other IIS phenotypes, and tissue-specific respiration was also increased through elevated mitochondrial biogenesis. Elevated mitochondrial biogenesis and respiration in the fat body were required for IIS-mediated longevity, and directly increasing mitochondrial biogenesis in the fat body extended life span. Expression of proteasomal subunits was modulated in the gut of the fly by reduced IIS. These changes resulted in increased proteasome assembly and activity, increased proteome maintenance and increased gut integrity with age. Inhibition of these changes abolished IIS-mediated longevity, and directly increasing gut-specific proteasome function extended life span. Importantly, we showed that these responses are pro-longevity and largely, tissue-specific. Due to the level of conservation of these processes, our study suggests the development of tissue-targeted pharmacological agents via homing peptides (Svensen *et al*, 2012) or chemical hybridization (Finan *et al*, 2016) to inhibit translation, activate the proteasome or increase respiration may be beneficial in prolonging longevity and health in mammals.

# Materials and Methods

### Fly stocks and fly husbandry

All mutants and transgenes were backcrossed into a white Dahomey ($w^{Dah}$) wild-type strain for at least eight generations. Fly stocks were kept at 25°C on a 12-h light and 12-h dark cycle and fed a standard sugar/yeast/agar diet (SYA; Bass *et al*, 2007). In all experiments, once-mated females were reared at controlled larval densities. Adult flies were aged (10 days) in SYA food vials (10–25 flies per vial) prior to analysis. Dissections were carried out in cold phosphate-buffered saline (PBS) and tissues either directly analysed or frozen on dry ice. Fly strains used in this study were as follows: *Tigs-Gene-switch (Tigs-GS)* (Buch *et al*, 2008; Poirier *et al*, 2008), *UAS-RPN6, UAS-rpr, InsP3-Gal4* (Buch *et al*, 2008; Slack *et al*, 2011), *UAS-Srl* (Tiefenböck *et al*, 2010), $dfoxo^{\Delta94}$ (Slack *et al*, 2011).

### Generation of UAS-RPN6 transgenic flies

To generate a transgenic fly stock for conditional over-expression of *Drosophila* Rpn6, the Rpn6 ORF was PCR amplified with primers SOL710 (ATGAATTCGCAAGATGGCCGGAGCAACAC) and SOL711 (ATGGTACCTTACGACAGCTTCTTAGCCTTC) and cDNA LD18931 as template (*Drosophila* Genomics Resource Center) and subsequently cloned into the pUAST attB vector using the EcoRI and KpnI restriction sites of the primers. Transgenic flies were generated using the φC31 and attP/attB integration system (Bischof *et al*, 2007), and transgenes were inserted into the attP40 landing-site.

### Peptide generation

Fly tissues (50/sample) from six biological replicates were lysed in pre-heated (95°C) 6M guanidine hydrochloride, 10 mM TCEP, 40 mM CAA, 100 mM Tris pH 8,5 lysis buffer. Following shaking at 18,407 *g* (95°C), tissues were sonicated for five cycles (Bioruptor plus). Lysis buffer was then diluted 11-fold in digestion buffer (25 mM Tris 8.5 pH, 10% acetyl nitride) and vortexed prior to a trypsin gold (Promega) overnight (37°C) digest (1 μg trypsin/50 μg protein). Samples were sonicated again for five cycles, prior to a further trypsin gold (Promega) overnight (37°C) digest (0.5 μg trypsin/50 μg protein) for 4 h (37°C) with gentle agitation. Samples were then placed in a SpeedVac (5 min, 37°C) to remove acetyl nitride. Peptides were desalted using SDB.XC Stage Tips (Rappsilber *et al*, 2003). Peptides were then eluted using (80% acetyl nitride, 0.1% formic acid) and placed in a SpeedVac (55 min, 29°C) to remove acetyl nitride, and quantified via Nanodrop.

### MS/MS

Peptides were loaded on a 50-cm column with 75 μm inner diameter, packed in-house with 1.8-μm C18 particles (Dr Maisch GmbH, Germany). Reversed phase chromatography was performed using the Thermo EASY-nLC 1000 with a binary buffer system consisting of 0.1% formic acid (buffer A) and 80% acetonitrile in 0.1% formic acid (buffer B). The peptides were separated by a segmented gradient of 0 to 20 to 40% buffer B in 0 to 85 to 140 min for a 160-min gradient run with a flow rate of 350 nl/min. The Q-Exactive was operated in the data-dependent mode with survey scans acquired at

a resolution of 120,000; the resolution of the MS/MS scans was set to 15,000. Up to the 20 most abundant isotope patterns with charge $\geq 2$ and $< 7$ from the survey scan were selected with an isolation window of 1.5 Th and fragmented by HCD (20) with normalized collision energies of 27. The maximum ion injection times for the survey scan and the MS/MS scans were 50 and 100 ms, respectively, and the AGC target value for the MS and MS/MS scan modes was set to 1E6 and 1E5, respectively. The MS AGC underfill ratio was set to 20% or higher. Repeat sequencing of peptides was kept to a minimum by dynamic exclusion of the sequenced peptides for 45 s.

### Protein identification and quantification

The .raw data files were analysed. Protein identification was carried out using MaxQuant (Cox & Mann, 2008) version 1.5.0.4 using the integrated Andromeda search engine (Cox *et al*, 2011). The data were searched against the canonical and isoform, Swiss-Prot and TrEMBL, Uniprot sequences corresponding to *Drosophila melanogaster* (20,987 entries). The database was automatically complemented with sequences of contaminating proteins by MaxQuant. For peptide identification, cysteine carbamidomethylation was set as "fixed" and methionine oxidation and protein N-terminal acetylation as "variable" modification. The *in-silico* digestion parameter of the search engine was set to "Trypsin/P", allowing for cleavage after lysine and arginine, also when followed by proline. The minimum number of peptides and razor peptides for protein identification was 1; the minimum number of unique peptides was 0. Protein and peptide identification was performed with FDR of 0.01. The "second peptide" option was on, allowing for the identification of co-fragmented peptides. In order to transfer identifications to non-sequenced peptides in the separate analyses, the option "Match between runs" was turned on using a "Match time window" of 0.5 min and "Alignment time window" of 20 min. Protein and peptide identifications were performed within, not across, tissue groups. Label-free quantification (LFQ) and normalization was done using MaxQuant (Cox *et al*, 2014). The default parameters for LFQ were used, except that the "LFQ min. ratio count" parameter was set to 1. Unique plus razor peptides were used for protein quantification. LFQ analysis was done separately on each tissue.

### Perseus informatics analysis

The results of the LFQ analyses were loaded into the Perseus statistical framework (http://www.perseus-framework.org/) version 1.4.1.2. Protein contaminants, reverse database identifications and proteins "Only identified by site" were removed, and LFQ intensity values were log2 transformed. After categorical annotation into four categories based on genotype ($dfoxo^{\Delta94}/w^{Dah}$) and ablation of mNSCs (yes/no), the data were filtered in order to contain a minimum of four valid values in at least one category. The remaining missing values were replaced, separately for each column, from normal distribution using width of 0.3 and down shift of 1.8. Based on quality control analysis, seven of the 90 samples (three fat body samples, three thorax samples and one brain sample) were excluded from the subsequent analysis due to technical failure of the chromatography. With those exceptions, the data were of high quality and reproducible between replicates. Information on Genotype and

    

tissue-specific average LFQ values and associated SEM values are included in Dataset EV1.

## Bioinformatics

PCA analysis was carried out using the FactoMineR R package (Lê *et al*, 2008). The LIMMA package (Bioconductor suite, R) was used for differential expression analysis (Schwämmle *et al*, 2013). Multiplicity-corrected *P*-values were calculated using the Benjamini-Hochberg procedure and significance determined using an adjusted *P*-value cut-off of 0.1.

Experimental groups were specified as follows: (A) $w^{Dah}$, (B) $dfoxo^{\Delta 94}$ mutant, (C) *InsP3-Gal4/UAS-rpr* and (D) *InsP3-Gal4/UAS-rpr*, $dfoxo^{\Delta 94}$. Hypotheses were tested as linear combinations against the null hypothesis of no change: ablation-induced changes in the wild-type background (C vs. A), ablation-induced changes in the $dfoxo^{\Delta 94}$ background (D vs. B) and their interaction term (C–A vs. D–B). The subset of proteins differentially expressed in C vs. A was further subdivided into two categories: Proteins that changed in response to mNSC ablation in a *dfoxo*-dependent manner, and those that behaved similarly regardless of genetic background were d*foxo*-independent: *Foxo-dependent:* Differential in C vs. A and the interaction term. If a protein was regulated in the same direction in C vs. A and D vs. B, it was required to show a stronger response in the first contrast. *Foxo-independent:* Differential in C vs. A. Protein expression levels were required to be equivalent with regard to differential expression in the interaction term. In any given tissue, a protein was identified as equivalent if the 95% CI of its log2 fold change fell within an interval ($[-t; t]$). The parameter $t$ was determined as minimum log-fold change of any significantly differentially expressed protein detected in the interaction term of each respective tissue: brain ($t = 0.072$), fat body ($t = 0.177$), gut ($t = 0.155$) and thorax ($t = 0.216$). Additionally, this subset included proteins found to be significant in the interaction term if they showed a stronger response to mNSC ablation in equal direction to that in the $dfoxo^{\Delta 94}$ background compared to the wild-type background.

## Binding site identification

We used FIMO, MEME suite (Bailey *et al*, 2009) to identify genes whose transcription start site was within 1,000 bp of a dFOXO binding motif, using the default *P*-value threshold ($1e^{-4}$). Sequences corresponding to genes of measured proteins were extracted from the BDGP6 reference genome, whilst the dFOXO binding motif was taken from Fly Factor Survey (Zhu *et al*, 2011). We then mapped FIMO hits to proteins whose expression changed under reduced IIS conditions or those that were identified as *dfoxo*-dependent and used a hypergeometric test to evaluate the significance of the overlap. Selection and background were limited to the set of genes corresponding to measured proteins.

## Gene ontology term enrichment

topGO was used for GO term enrichment analysis and gene annotation performed using the Bioconductor annotation package org. dm.e.g.db. To identify enriched GO terms, the one-sided elim Fisher procedure was used ($\alpha \leq 0.05$; Alexa *et al*, 2006). The enrichment score of a GO term is defined as *log*2 (#*Detected significant genes*/

#*Expected significant genes*). For individual tissue-specific contrasts, the subsets of differentially expressed proteins were tested for term enrichment against a tissue-specific reference background. Cell plots show the most specific significantly enriched categories with a minimum of five significant associated proteins. To functionally characterize the individual tissues of wild-type flies, GO enrichment analysis was carried out for proteins detected exclusively in the respective tissue, against the background of all detected proteins. For PCA map functional annotation (Fig 1D), the top 5% and lowest 5% quantiles of proteins according to their contributions to each dimension were tested for GO term enrichment, against the background of all detected proteins. The plot shows the 10 most significant terms with a minimum of 10 significant proteins, for each direction of the two dimensions.

## Network propagation

Network propagation (Vanunu *et al*, 2010) was done using the *D. melanogaster* PPI network DroID (Murali *et al*, 2011) filtered for high confidence edges (40% confidence, $n = 34,866$). The network was converted to an adjacency matrix and normalized with the Laplacian transformation (using the graph.laplacian function in the igraph R package). Differentially regulated proteins were classified into *dfoxo*-dependent and *dfoxo*-independent (see Bioinformatics section above). For the analysis of *dfoxo*-dependent proteins, the *P*-values of proteins not belonging to this group were excluded (set to 1). Likewise, for the evaluation of the *dfoxo*-independent set, *P*-values of proteins not detected as such were excluded. Finally, the *P*-values were −log2 transformed and mapped to the network. After mapping, the transformed *P*-values on the individual nodes were diffused to their adjacent nodes using the spreading coefficient of 0.8 (corresponds to the percentage of sharing to neighbours). This spreading changed the scores of all proteins in the network, and it was iteratively repeated until protein scores do not change anymore.

We noticed that the network topology (i.e. the network structure) induces a bias: certain regions of the network tend to accumulate high scores during the propagation even if there is no actual signal. In order to correct for this bias, we computed a node-specific topology bias in the following way: we started with equal scores on each node (e.g. each node has a score of 1). Then, we perform the propagation as described above. The resulting scores reflect the bias and are subtracted from the node scores after propagating the actual node scores. Subsequently, the corrected propagated scores were clustered based on their Euclidean distances (using the hclust function in R) allowing for the identification of enriched clusters for each tissue. For visualizing the enriched clusters from *dfoxo*-dependent and *dfoxo*-independent proteins on the same heatmap, the propagated profiles were subtracted (*dfoxo*-dependent smoothed scores and *dfoxo*-independent smoothed scores). GO enrichment analysis of individual clusters was done using the topGO R package defining all proteins that are part of the network as background.

## $^{35}$S-methionine/$^{35}$S-cysteine incorporation assay

Incorporation assays were performed as previously described. Briefly, 5–8 tissues were dissected and collected in DMEM

(#41965-047, Gibco). For labelling, DMEM was replaced with methionine and cysteine free DMEM (#21-013-24, Gibco), supplemented with $^{35}$S-labelled methionine and cysteine (#NEG772, Perkin-Elmer) and incubated in uncapped Eppendorf tubes on a shaking platform at room temperature (60 min). Samples were then washed in ice cold PBS and lysed in RIPA buffer (150 mM sodium chloride, 1.0% NP-40, 0.5% sodium deoxycholate, 0.1% SDS, 50 mM Tris, pH 8.0). Lysates were cleared by centrifugation at 15,871 $g$, 4°C for 10 min. Protein was precipitated by adding 1 volume of 20% TCA, incubating for 15 min on ice and centrifugation at 15,871 $g$, 4°C for 15 min. The pellet was washed twice in acetone and resuspended in 4 M guanidine HCl. Half the sample was added to 10 ml of scintillation fluid (Ultima Gold, Perkin-Elmer) and counted for 5 min per sample in a scintillation counter (Perkin-Elmer). The remaining sample was used to determine protein content using BCA (Pierce), following the manufacturers protocol. Scintillation counts were then normalized to total protein content prior to statistical analysis.

### Respiratory rate measurements

Flies (10 days) were dissected in PBS and transferred to respiratory buffer (120 mM sucrose, 50 mM KCl, 20 mM Tris–HCl, 4 mM $KH_2PO_4$, 2 mM $MgCl_2$, 1 mM EGTA, 0.01% digitonin, 0.05% BSA, pH 7.2). Oxygen consumption was measured using an oxygraph chamber (OROBOROS) at 25°C. Complex I-dependent respiration was assessed using the substrates proline (10 mM), pyruvate (10 mM), malate (5 mM) and glutamate (5 mM), along with ADP (1.25 mM). The respiration was uncoupled by the addition of CCCP (0.3 μM). Maximum flux was then measured by adding complex II substrates succinate (10 mM) and glycerol-3-phosphate (5 mM). Rotenone-sensitive flux was measured in the presence of rotenone (3 μM). Finally, dry weights of tissues were determined to normalize the oxygen consumption.

### 20s proteasome activity assay

Proteasome caspase-like activity was measured as previously described (Vernace *et al*, 2007). Briefly, tissues were dissected in PBS and homogenized in 25 mM Tris (pH 7.5), and debris cleared by centrifugation. Protein content was measured, and 15 μg used to measure activity by incubating with 12.5 μM Z-Leu-Leu-Glu-AMC (Enzo Life Sciences) at a total volume of 200 μl. AMC fluorescence was measured at 360 nM excitation and 460 nM emission using a spectrofluorimeter (Tecan). Free AMC was used as a standard every 2 min for 30 min.

### In-gel proteasome assembly/activity assay

In-gel activity/assembly assay was performed as previously described (Vernace *et al*, 2007). Briefly, individual tissues were dissected in PBS and homogenized in proteasome buffer (50 mM Tris–HCl, pH 7.4, 5 mM $MgCl_2$, 1 mM ATP, 1 mM DTT and 10% glycerol) on ice, centrifuged at 16,000 $g$ (4°C, 10 min) and equal amounts of proteins loaded on a Bio-Rad TGX 7.5% precast gel. Gels were incubated in proteasome buffer containing 0.4 mM Z-Leu-Leu-Glu-AMC (Enzo Life Sciences) for 15 min (37°C). Proteasome bands were visualized with UV light (360 nm) on a ChemiDoc station

(Bio-Rad). Differential activity was estimate by densitometry using ImageJ.

### Western blotting

Western blots were carried out on protein extracts of individual dissected tissues. Proteins were quantified using BCA (Pierce), and equal amounts loaded on Any-KD pre-stained SDS–PAGE gels (Bio-Rad) and blotted according to standard protocols. Antibody dilutions were as follows: K48-poly Ubiquitin (Cell Signaling Technology D9D5 #8081; 1:1,000), Rpt6 (Santa Cruz biotechnology 9E3 #65752; 1:100), NDUFS3 (Abcam 17D95 #14711; 1:1,000). Appropriate secondary antibodies conjugated to horseradish peroxidase were used at a dilution of 1:10,000.

### Life span analysis

Once-mated flies, reared at standard densities, were transferred to vials (10-25/vial). Flies were transferred to fresh vials three times a week, and deaths scored on transferal. Standard SYA food (Bass *et al*, 2007) was used throughout but, where needed, RU486, EtOH (vehicle control) was added to the food at a final concentration of 200 μM, or for bortezomib at 1–3 μm.

### Gut integrity assay ("Smurf" assay)

Flies were grown and aged on standard SYA food until 48 h prior to the assay and then switched to SYA food containing 2.5% (w/v) Brilliant Blue FCF (AppliChem). Flies were then examined for "smurfing" and were recorded as "smurf" or not "smurf", as previously described (Rera *et al*, 2012; Vizcaíno *et al*, 2014), except flies were kept on dyed food for 48 h.

### Quantitative real-time PCR

Total RNA was extracted using Trizol (Invitrogen Corp.) according to the manufacturer's instructions, including a DNase treatment. cDNA was prepared using SuperScript III first-strand synthesis kit (Invitrogen Corp.). Quantitative real-time PCR was performed in a 7900HT real-time PCR system (Applied Biosystems). Relative expression (fold induction) was calculated using the $\Delta\Delta C_T$ method and Rpl32 or *Actin5c* as a normalization control.

### Statistical analysis

Statistical analysis was performed using Graphpad Prism and JMP. Individual statistical tests are mentioned in appropriate figure legends. Lifespan assays were recorded using Excel and survival was analysed using log rank test.

### Data availability

The mass spectrometry proteomics data from this publication have been deposited to the ProteomeXchange Consortium (Vizcaíno *et al*, 2014) via the PRIDE archive database (Vizcaíno *et al*, 2016) https://www.ebi.ac.uk/pride/archive/ and assigned the accession PXD006225.

**Expanded View** for this article is available online.

## Acknowledgements

Stocks from the Bloomington Drosophila Stock Center (NIH P40OD018537) were used in this study. We would also like to acknowledge Ilian Atanassov and the Proteomic facility at the Max Planck Institute for Biology of Ageing for help during MS analysis. We acknowledge funding from the Max Planck Society (to LT, RS, CJ, MR, SG, JF, NN, MM, LP), a Wellcome Trust Strategic Award (WT098565/Z/12/Z) (to LP), the European Union's Framework Programme for Research and Innovation Horizon (2014–2020) under the Marie Sklodowska-Curie Grant Agreement no. 655623 (to PE), a Bundesministerium für Bildung und Forschung Grant SyBACol 0315893A-B (to AB, CD, MC, LT, LP). We also acknowledge funding form Biotechnology and Biological Sciences Research Council (NA: BB/M029093/1) (to NA), the Medical Research Council (NA: MR/L018802/1) (to NA) and the Royal Society (NA: RG140694) (to NA). CD was supported by the Klaus Tschira Stiftung gGmbH [00.219b.2013]. The research leading to these results has received funding from the European Research Council under the European Union's Seventh Framework Programme (FP7/2007-2013)/ERC grant agreement number 268739 (to LP).

## Author contributions

LST, LP designed the experiments. LST, CJ, NN, PE, MR, SG, and JF performed the experiments. LST, RS, MC, NN, CD, MM, NA, AB, and LP designed the data analysis. LST, RS, and MC performed the analysis. LST, NA, and LP wrote the manuscript.

## Conflict of interest

The authors declare that they have no conflict of interest.

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
