## [Review Process File · Molecular Systems Biology]

A proteomic atlas of insulin signaling reveals tissue-specific mechanisms of longevity-assurance

Luke S. Tain, Robert Sehlke, Chirag Jain, Manopriya Chokkalingam, Nagara Nagarajuna, Paul Essers, Mark Rassner, Sebastian Grönke, Jenny Froelich, Christoph Dieterich, Matthias Mann, Nazif Alic, Andreas Beyer & Linda Partridge

Corresponding authors: Andreas Beyer, CECAD & Linda Partridge, Max-Planck Institute for Biology of Ageing

Review timeline:	Submission date:	31 March 2017
	Editorial Decision:	28 April 2017
	Revision received:	24 July 2017
	Editorial Decision:	09 August 2017
	Revision received:	15 August 2017
	Accepted:	16 August 2017

Editor: Maria Polychronidou

Transaction Report:

1st Editorial Decision

28 April 2017

Thank you again for submitting your work to Molecular Systems Biology. We have now heard back from the three referees who agreed to evaluate your study. As you will see below, the reviewers appreciate that the topic of the study is interesting. However, as you will see below, they raise several significant concerns, which preclude the publication of the study in its current form.

Since most of the issues raised are related to the quality of the data and the conclusiveness of the main findings, they need to be convincingly addressed in a major revision. Please note that our editorial policy in principle allows a single round of major revision.

Without repeating all the specific points listed below, some of the more fundamental issues are the following:

- Reviewer #2 points out that the quality of the proteomics data and the validity of the conclusions derived from them needs to be better supported. Since these datasets are integral to the findings reported in the study, it is very important that their quality is convincingly demonstrated.
- Reviewer #3 mentions that further analyses are required to better support several of the conclusions reported in the study and provides constructive suggestions in this regard.

- During our pre-decision cross-commenting process, in which the referees are given the chance to comment on each other's reports, reviewer #3 agreed with the issues raised by reviewer #1 but mentioned that in his/her opinion, these issues (i.e. that some phrasing can be improved, that more synthesis of the results would improve the manuscript and that the term 'aging' should be replaced with lifespan') can be addressed in a revision, since they do not undermine the main conclusions.

REVIEWER REPORTS

Reviewer #1:

Review

"A Proteomic Atlas of Insulin Signaling Reveals Tissue-specific Modulation of Metabolism and Proteostasis to Ameliorate Ageing."
MSB-17-7663

In this manuscript, Tain et al. performed proteomic studies of four insulin-sensitive tissues in an insulin/IGF-1 long-lived *Drosophila* mutant. Overall, this is an interesting study. However, the manuscript is extremely difficult follow. There are numerous grammatical errors, Long sentences, etc. that make it extremely difficult for the reader to follow the scientific results and experiments. In addition, there is a clear lack of synthesis of the findings. The experiments do not support many of the bold conclusions in the manuscript. For example, the manuscript focuses on lifespan which is clearly not the same as aging. Finally, the manuscript begins with a tissue-specific proteomic analysis,. Then, the authors state they use a network approach followed by analysis on individual genes in each tissue. There is little synthesis of the results in the different tissues and how they would coordinate to promote longevity. Therefore, at this time, the manuscript should be rejected.

Major points

1) 44% of the predicted proteome was identified. Indicate whether this is similar compared to other studies. How does this influence your findings? Especially since you suggest you are trying to obtain "systems-level insight "? How did "systems-level insight " advance this study? How was the network propagation approach used and how did it help advance the findings?

2) The manuscript needs to be editorially revised. It is extremely difficult to read, follow the experiments, results and overall synthesis of the findings. Below are a few examples:
Since there are no page numbers, this reviewer can't tell you where the sentences are from.

a) "We took the set of proteins that changed expression with ablation of mNSCs in wild type flies (which proteins?) and asked whether their response to mNSC-ablation was different in a dfoxo-null background, by profiling the tissue-specific proteomes of the mNSC-ablated flies lacking dfoxo (*Insp3-Gal4/UAS-rpr; dfoxoΔ/Δ*) and the corresponding *dfoxoΔ/Δ* controls."
-4 line sentence too

b) a concluding paragraph: Long sentences, hard to understand the concept, and not sure the data actually supports these ideas.

"Overall, our data reveal that lowered systemic IIS coopts different mechanisms to regulate respiratory state (?) reciprocally in different tissues decreasing respiration in the gut independently of dfoxo, whilst simultaneously increasing it (what is it?) in the fat body in a dfoxo- and spargel/delg-dependent manner. Increasing respiration, through increased mitochondrial biogenesis in the fat body of *Drosophila* is sufficient to extend lifespan, indicating that this tissue specific response to reduced IIS at least partially mediates the extended lifespan of IIS mutants." What does this mean?

c) "Together, our data suggests a tissue-specific, dfoxo-dependent regulation of proteasome function

in response to reduced IIS that correlates with the longevity phenotype. "???"

d) Why are the dilp2-3,5 mutants used in experiments such as At a concentration of 2 μ M Bortezomib did not reduce the lifespan of wild type flies (Fig. 6A), but significantly reduced the lifespan-extension in mNSC-ablated flies (Fig. 6A), and independently, in dilp2-3,5 mutants (Fig. S4A). " ??

e) at least three times in the manuscript, the words our data suggest are used. However there is little integration of the findings.

3) Overall, the conclusions are not supported by the results. It is unclear how the network propagation approach guided the studies. The manuscript seems to read like two separate papers. The first with the systems analysis using proteomics, and the second a molecular genetic analysis of genes in different tissues. After reading the manuscript, this reviewer isn't clear how this manuscript helps to understand the mechanism underlying long-lived *Drosophila* IIS mutant.

Minor Points

-please make sure all abbreviations are explained.

-Page numbers would help.

Reviewer #2:

This study aims to identify the tissue-specific response of insulin signaling in *Drosophila* in order to understand the impact of this signaling pathway on longevity. To this end, the authors use two model systems to blunt insulin signaling resulting in increased lifespan, one ablating neurosecretory cells which secrete insulin-like peptides, and the other knocking out Foxo, a transcription factor required for insulin signaling. They then take a proteomic approach to profile proteome composition in 4 different tissues (brain, gut, fat body and muscle) in wt and mutant flies, identifying many hundreds of proteins whose expression depends on insulin signaling in a tissue-specific manner. These proteins are grouped in functional entities by crossing the data with protein interaction data, the results which were validated in a series of dedicated functional assays. This demonstrated that proteasomal, mitochondrial and ribosomal activity are controlled by insulin signaling in a tissue-specific manner.

This is an interesting study demonstrating that the effects of insulin signaling is highly context (tissue-type) dependent, leading to the novel hypothesis that the specific responses across the various tissues collectively contribute to longevity. While the functional studies leading to this conclusion are largely sound, the proteomic data look less convincing requiring further elaboration and clarification to demonstrate robustness of the data, and validity of the overall conclusions deduced from them:

1. A main concern is the lack of detail with regard to protein quantification, and the quality of quantitative data. In particular, the authors claim that a surprisingly large number of protein expression changes upon blunting insulin signaling, however the details are missing to show this convincingly. For instance, out of ~4000 proteins identified in brain tissue, 1300 are said to change, however without mentioning the magnitude of these effects. A quick view on the data presented in table S3 indicates that out of these 1300 proteins, 1000 change less than 2-fold, and ~500 even less than 1.4-fold (0.5 on log₂ scale). This is a suspiciously small difference that can usually only be obtained for highly controlled and homogeneous systems, which is not the case here employing label-free quantification in an *in vivo* system. The large proportion of differentially expressed proteins may well be explained by performing only few replicate experiments (number not stated in

the manuscript, should be added) while allowing a relatively large FDR (10%) resulting in many proteins with a relatively small change in expression. The least the authors should do for each tissue is to show the correlation between fold change and p-values in a volcano plot, to emphasize that many proteins change only marginally and with a relatively low confidence level. Even then, it may not be surprising that '60% of these of these proteins were not previously identified as regulated by reduced IIS' (page 6), and the claim that 'substantial, new information was gained' remains a leap of faith.

2. On a technical note, protein identification in this study seems to heavily rely on the 'match between runs' option in MaxQuant. In fact it is unclear why it was used here considering the profound differences in proteome composition between tissues - a scenario for which this option was explicitly NOT designed. Since this introduces the risk of introducing erroneous transfer of peptide identifications, the authors should indicate the FDR of protein assignment between disparate tissues.

3. Page 7: by comparing proteomic data sets obtained from both biological model systems, the authors identify 361 and 196 proteins whose expression do and do not depend on Foxo, respectively. However, in light of point 1 above, since the number of proteins deemed to change in expression is likely to be inflated, the robustness of this analysis can be questioned. Details should be added on the effect size in protein expression, and the error in these values taking into account error propagation from the respective proteomic data that are compared.

Reviewer #3:

This is a nice study with new insights into the regulation of lifespan by Insulin/IGF Signaling (IIS). Using a mass spectrometry tour de force, Tain et al. define the changes in the proteome of the fly gut, brain, fat body and muscle upon reduction of organismal Insulin levels in both control and foxo mutant flies. This allows them to identify which changes in the proteome occur in a foxo dependent or independent manner. Interestingly, they find that different tissues in the fly change their proteome in different ways in response to reduced IIS.

Tain et al also test the contribution of altered mitochondrial activity and proteasomal activity in the fat body and gut, respectively, on aging. This part is also nice because it shows that elevated respiration in the fat body and elevated proteasomal activity in the gut both contribute towards the extended lifespan of NSC-ablated flies.

Several links remain unexplored. For instance, it is not clear how FOXO regulates mitochondrial activity or proteasomal function (ie upstream), nor is it clear how these two altered functions affect lifespan (ie downstream). However this manuscript serves as a good starting point for future studies on these two topics, and the identification of these two processes is already an important contribution to the field.

Although most of the data are solid, some major issues need to be addressed to make the main claims of the paper solid:

1. The authors rightly claim that the lifespan-relevant changes should be occurring downstream of FOXO. Since FOXO is a transcription factor, it affects mRNA levels. Yet the effects described here are occurring at the protein level. Hence there is a disconnect between the FOXO-dependent transcriptional changes and the proteome changes. The authors should test whether the main proteins discussed in this manuscript, which change in abundance upon reduced IIS, are FOXO targets. Are the 6 mitochondrial proteins shown in Figure 4A, which increase in the fat body, also up-regulated at the mRNA level in a foxo dependent manner? ie are these bona-fide foxo targets or do they change in abundance as a secondary consequence of changes in foxo targets? Likewise, the mRNA levels for the proteasomal subunits highlighted in yellow in Fig 5A should be quantified and presented (especially Rpt6).

2. In Fig 4B, the quantification does not seem to correspond to the image of the western blot. In the WB, NDUF3 protein levels appear reduced in the foxo94 lysates compared to wDah (despite higher loading control levels) yet the quantification does not show this. If NDUF3 is not the best protein to show these changes, then another protein should be shown. In any case, the changes in

respiration are unlikely to be due to only one protein changing in abundance.

3. In Figure 4F, are spargel protein and mRNA levels in the fat body actually affected by IIS-ablation and are those changes foxo-dependent? If not, this should be clearly stated. Otherwise the text gives the impression that regulation of mitochondrial activity in response to reduced IIS goes via Spargel regulation.

4. Are the changes in Rpt6 levels shown in Figure 5B also occurring in a foxo mutant background? (ie are they foxo dependent?). This should be tested at the mRNA and protein levels.

5. Regarding Figure 5, the authors conclude: "These data suggest that reduced IIS results in increased proteasomal assembly in the gut, through increased levels of Rpt6". There do not seem to be any data showing that the increased proteasomal assembly is due to increased levels of Rpt6. Only correlative data are shown. Either data should be provided, or the claim should be fixed.

Minor issues:

1. Have the authors tested if the discovered changes in translation rates in the fat body are linked to lifespan? This should be mentioned/discussed in the text.

2. Figure S2: With the exception of the fat body, the other organs show a very high variability in the S35-incorporation and this does not seem to correspond to the size of the error bars in the quantifications. What are the error bars showing? The figure legends do not indicate if the error bars are std. dev. and over how many biological replicates. This information should be added, and perhaps the error bars re-calculated.

3. Figure 4C-D: the respiration assays are missing from the materials & methods.

4. The authors write "Furthermore, expression of Spargel or delg in the fat body of dilp2-3,5 mutants was required for longevity but did not affect the lifespan of wild type controls (Fig. 4F, S3D-E) " but this conclusion seems to strong given the provided data. Spargel is not required for the increase in lifespan upon IIS-ablation, but it is contributing to it as the lifespan of Spargel knockdown animals still can be significantly prolonged by IIS-ablation.

5. Figure 6I - The lifespan effect is very minor. Rephrase as "small but significant"?

6. Wrong figure reference: "Adult-specific over-expression of RPN6 was sufficient to increase proteasome activity, increase gut integrity and extend lifespan (Fig 6I, Fig S4A-C)." Should be "(Fig 6I, Fig S5A-C)" ?

7. The protein expression data will be of great value to the community. The authors should summarize in a supplemental table (excel? csv?) these data, showing protein expression values (or peptide counts?) in the four tissues for the various genotypes (+/- IIS, +/- FOXO).

1st Revision - authors' response

24 July 2017

Reviewer #1:

Major points

Point 1. 44% of the predicted proteome was identified. Indicate whether this is similar compared to other studies. How does this influence your findings? Especially since you suggest you are trying to obtain "systems-level insight "? How did "systems-level insight " advance this study? How was the network propagation approach used and how did it help advance the findings?

Author response

The reviewer has made several different, but linked, points.

First, the reviewer asks how our study compares to similar studies. There are very few comparable proteomic studies in Drosophila. In a recent study by Aradska et al., 4613 proteins were identified in Drosophila heads, comprising 33% of the predicted proteome (Aradska et al, 2015). Our analysis

represents a considerable increase in coverage compared to the Aradska study, identifying 6085 proteins, 44% of the predicted proteome.

Second, the reviewer asks how the 'systems level approach' advanced this study? Our study focuses on the proteome response to reduced IIS and, for the first time, uses dfoxo-dependency as a means to identify proteomic changes that are specifically associated to longevity. Through quantifying thousands of proteins and through studying the networks they are involved in we move away from single-gene and single-protein analysis. Additionally, we assess expression at the fly system level. Thus, for the first time we have simultaneously examined the tissue-specific proteomes of the fly, allowing us to monitor tissue-specific effects of insulin signaling. This systems-level analysis advanced the study, because it allows us to study multiple molecular and cellular processes in response to insulin signaling. For example, through this analysis it became possible to distinguish foxo-dependent from foxo-independent processes.

Third, the reviewer asks how the network propagation approach was used and how it advanced the findings. Despite recent improvements, MS-based shotgun proteomics does not cover all proteins. In addition, measurements are affected by technical and biological noise. In order to account for these problems and in order to aid the molecular interpretation of the results we have used network propagation. Unlike traditional GO enrichment analyses, network propagation does not rely on predefined annotations. For example, using network propagation, it is possible to find sub-networks connecting two molecular pathways or bridging different functions. In this study, network propagation helped us to identify additional proteins of interest even if they were not quantified in the experiments and helped to 'de-noise' the data. In particular it identified or enriched relevant cellular functions, such as the proteasome complex or mitochondrial respiration, that were either only weakly or not detected with traditional differential expression analysis.

Network propagation aids in inferring the protein sub-networks that are altered in response to reduced IIS. It starts with a pre-defined network (in our case DroID (Murali et al, 2011)) and a set of quantified nodes, in our case the proteins that are differentially regulated in a foxo-dependent/independent manner. The initial negative log-transformed p-values of regulated proteins were then propagated to the neighboring nodes in the network – separately for each tissue. The spreading of scores was done iteratively until the scores no longer changed, as previously described in (Vanunu et al, 2010). To make this clear we have edited the Network propagation description in the method section. To illustrate the network propagation algorithm we have included Responses Figure 1. This shows the network propagation on a local scale, specifically on the proteasome complex in the gut. Here, few nodes (proteasomal sub-units) of the complex were initially quantified as dfoxo-dependently regulated in response to reduced IIS. Using network propagation we were able to observe enrichment of the proteasome complex in the Gut tissue. This led us to experimentally validate the suggested tissue-specific, dfoxo-dependent responses to reduced IIS as described in the main text (pg 12) and shown in Figure 5A-E.

Responses Figure 1. Network propagation, specifically on the proteasome complex in the gut. The proteasome sub-network initial scores (left) and propagated scores (right) are shown for the 33 proteasomal subunits in Drosophila. Node colors denote $-\log_2(\text{p-value})$ on the left and network propagated scores on the right.

To illustrate the improvements in results through network propagation, we have shown the enrichment of the processes that are specifically enhanced by propagation in a tissue specific way (Responses Figure 2). These are the three examples that were identified through network propagation and they were all experimentally validated for their tissue-specific involvement in determination of lifespan.

Responses Figure 2: Functional enrichment of proteins that are either differentially regulated in reduced IIS or identified through network propagation. The length of the bars corresponds to \log_2 (significant no. of proteins / expected no. of proteins). The asterisks at the ends of bars indicate statistical significance for the respective GO terms from the topGO elim Fisher's exact test ($p < 0.05$; ** $p < 0.01$; *** $p < 0.001$; **** $p < 0.0001$).*

Point 2A.

"We took the set of proteins that changed expression with ablation of mNSCs in wild type flies (which proteins?) and asked whether their response to mNSC-ablation was different in a dfoxo-null background, by profiling the tissue-specific proteomes of the mNSC-ablated flies lacking dfoxo (Insp3-Gal4/UAS-rpr; dfoxo Δ/Δ) and the corresponding dfoxo Δ/Δ controls."

-4 line sentence too

Author response

We thank the reviewer for pointing out an overly long sentence and have broken it into two. The revised sentences are shown below and have been inserted into the text on pg.7.

"We took the set of proteins that changed expression with ablation of mNSCs in wild type flies and asked whether their response to mNSC-ablation was abrogated in a dfoxo-null background.

Accordingly, we additionally profiled the tissue-specific proteomes of the mNSC-ablated flies lacking dfoxo (Insp3-Gal4/UAS-rpr; dfoxo Δ/Δ) and the corresponding dfoxo Δ/Δ controls."

Point 2B concluding paragraph: Long sentences, hard to understand the concept, and not sure the data actually supports these ideas.

"Overall, our data reveal that lowered systemic IIS coopts different mechanisms to regulate respiratory state (?) reciprocally in different tissues decreasing respiration in the gut independently of dfoxo, whilst simultaneously increasing it (what is it?) in the fat body in a dfoxo- and spargel/delg-dependent manner. Increasing respiration, through increased mitochondrial biogenesis in the fat body of *Drosophila* is sufficient to extend lifespan, indicating that this tissue specific response to reduced IIS at least partially mediates the extended lifespan of IIS mutants." What does this mean?

Author response

We thank the reviewer for highlighting two overly long and unclear sentences. We have edited the text to both shorten and clarify our points, this edit is shown below and is now inserted on pg 11 of the MS.

“Overall, our data reveal that lowered systemic IIS coopts different mechanisms to regulate respiration in different tissues. Reduced IIS decreased respiration in the gut independently of dfoxo, whilst simultaneously increasing respiration in the fat body in a dfoxo- and spargel/delg-dependent manner. Furthermore, increased mitochondrial biogenesis, and thus respiration, in the fat body is both necessary and sufficient to extend lifespan.”

Point 2C "Together, our data suggests a tissue-specific, dfoxo-dependent regulation of proteasome function in response to reduced IIS that correlates with the longevity phenotype. "???"

Author response

We have edited text to clarify this statement. The edited version is shown on pg. 12 and below. “Thus, increased proteasome activity in the gut in response to reduced IIS is a candidate mechanism for increased longevity.”

Point 2D Why are the dilp2-3,5 mutants used in experiments such as At a concentration of 2 μ M Bortezomib did not reduce the lifespan of wild type flies (Fig. 6A), but significantly reduced the lifespan-extension in mNSC-ablated flies (Fig. 6A), and independently, in dilp2-3,5 mutants (Fig. S4A). " ??

Author response

The reviewer asks why we have used dilp2-3,5 mutants alongside mNSC-ablated flies in figure 6A and figure S4A.

We have used dilp2-3,5 mutants, alongside mNSC-ablated flies, several times throughout the manuscript to establish the generality of the response to reduced IIS, as mentioned in the main text on pg. 12. In this example the extended lifespan of both IIS mutants is shortened by treatment with the proteasomal inhibitor bortezomib.

Point 2E At least three times in the manuscript, the words our data suggest are used. However there is little integration of the findings.

Author response

We have used the phrase ‘our data suggests’ three times in the manuscript once on page 12, 13, and once on page 15. These sentences have been edited (see below) to prevent repetition of similar phrasing within the MS.

Pg. 12 - “Together, our data suggests a tissue-specific, dfoxo-dependent regulation of proteasome function in response to reduced IIS that correlates with the longevity phenotype.“

This sentence has been edited in response to point 2C. The edited version is shown below and on pg. 12 of the MS.

“Thus, increasing proteasome function in the gut in response to reduced IIS correlates with longevity.”

Pg. 13 - “Our data suggest that increased proteasomal assembly/activity, and thus clearance of K48 poly-ubiquitinated proteins, in reduced IIS flies may underlie the IIS-mediated longevity through enhancing proteome maintenance.”

This sentence has been edited to prevent repetitive use of the phrase ‘our data suggest’. The edited sentence has been incorporated into the MS on pg. 13, and is shown below.

“Therefore, increased proteasomal assembly/activity, and thus clearance of K48 poly-ubiquitinated proteins, may underlie IIS-mediated longevity through enhanced proteome maintenance.”

Pg. 15 – “Together, our data suggests that increased levels of RPN6 are sufficient to increase proteasome assembly and activity, resulting in increased clearance of K48 poly-ubiquitinated proteins, enhancing proteome maintenance, and thus the health of the gut.”

This sentence concludes a section in the MS that shows that gut-specific over expression of RPN6 is sufficient to increase proteasomal assembly and activity, reduce K48-Ub protein levels, improve gut integrity and extend lifespan (Figure 6D-I). As such we believe the current sentence is justified. However, we have rephrased the sentence, exchanging ‘suggests’ to ‘show’ and inserted it into the MS on pg. 15 and below.

“Thus increased levels of RPN6 are sufficient to increase proteasome assembly and activity, and increased clearance of K48 poly-ubiquitinated proteins, enhancing proteome maintenance, and thus the health of the gut.”

Point 3 Overall, the conclusions are not supported by the results. It is unclear how the network propagation approach guided the studies. The manuscript seems to read like two separate papers. The first with the systems analysis using proteomics, and the second a molecular genetic analysis of genes in different tissues. After reading the manuscript, this reviewer isn't clear how this manuscript helps to understand the mechanism underlying long-lived *Drosophila* IIS mutant.

Author response

The reviewer has made several different points.

First, the reviewer believes the conclusions are not supported by the results. We have edited the text where suggested by the reviewer. We consider that these edits helped clarify the link between results and the conclusions drawn for the crucial points.

Second, the reviewer states that it is unclear how the network propagation approach guided the studies. In our response to point #1, and additionally in Responses figure 1&2, we have clarified the use of network propagation and showed how it directed our studies. To ensure the use of the method is transparent we have also edited the description of network propagation in the methods.

Third, the reviewer states “The manuscript seems to read like two separate papers. The first with the systems analysis using proteomics, and the second a molecular genetic analysis of genes in different tissues.”

In contrast to the reviewer we think that the combination of system-level tissue-specific proteomic analysis and functional in vivo follow up studies is one of the major strengths of this manuscript, which makes it stand out from many manuscripts that are purely based on descriptive high-throughput data. Recently, publications that include large scale datasets, such as the one presented here, have adopted this style, including publications within Molecular Systems Biology (Stout et al, 2013; Narayan et al, 2016; Ori et al, 2015) ...

Minor Points

-please make sure all abbreviations are explained.

Author response

We have ensured that the abbreviations meet the requirements of Molecular Systems Biology

-Page numbers would help.

Author response

We have now included page numbers.

Additional point

Author response

In their initial summary of the manuscript the reviewer suggests our study lacks synthesis between the findings. We have edited the text to include discussion on how alterations to tissue-specific proteomes and processes, in response to reduced IIS, may result in a coordinated pro-longevity

response. This text has been added to the MS, within the discussion on pg. 20, and below. Furthermore, we have added a summary figure (Figure 7), which shows a graphical synthesis about the individual, tissue-specific, dfoxo-dependent responses to reduced IIS.

“A system-level approach to analyze the tissue-specific proteome response to reduced IIS, followed by functional genetic and molecular analysis, has allowed the identification of IIS-mediated, tissue-specific pro-longevity processes. Despite their highly specific responses to lowered IIS, individual tissues each contributed to the increase in lifespan in the IIS mutant flies. As well as its effect on ageing, IIS co-ordinates tissue-specific processes to regulate early life traits such as development, growth and reproduction in response to nutrition and other cues. The level of IIS appears to have evolved to optimise these early life fitness traits, but evidently this level is too high for optimal health at later ages not normally subject to natural selection, resulting in pro-longevity effects of reduced IIS.

Here, we focused on two pro-longevity effects, gut-specific proteasome assembly/activity and fat body-specific respiration/mitochondrial biogenesis. Increasing proteasome activity in the gut was sufficient to extend lifespan by 11%, and elevating mitochondrial respiration in the fat body can extend lifespan to the same extent (11%). However, reducing IIS extends lifespan to a greater extent. For example mNSCs-ablated flies are approximately 30% longer lived than controls. Thus, it is tempting to suggest that increasing both proteasome activity and mitochondrial respiration in the same fly could lead to an additive increase in lifespan, and that manipulation of other processes identified here, such as reduced translation in the fat body, may extend lifespan even further (Fig. 7), a hypothesis that should be tested in future experiments.”

Figure 7. Model of tissue-specific responses to reduced IIS that may mediate longevity. Colours denote different tissues (fat body – orange, gut – purple). Dashed lines show tissue-specific responses that are both necessary and sufficient to extend life span.

Reviewer #2:

Point 1. A main concern is the lack of detail with regard to protein quantification, and the quality of quantitative data. In particular, the authors claim that a surprisingly large number of protein expression changes upon blunting insulin signaling, however the details are missing to show this convincingly. For instance, out of ~4000 proteins identified in brain tissue, 1300 are said to change,

however without mentioning the magnitude of these effects. A quick view on the data presented in table S3 indicates that out of these 1300 proteins, 1000 change less than 2-fold, and ~500 even less than 1.4-fold (0.5 on log₂ scale). This is a suspiciously small difference that can usually only be obtained for highly controlled and homogeneous systems, which is not the case here employing label-free quantification in an in vivo system. The large proportion of differentially expressed proteins may well be explained by performing only few replicate experiments (number not stated in the manuscript, should be added) while allowing a relatively large FDR (10%) resulting in many proteins with a relatively small change in expression. The least the authors should do for each tissue is to show the correlation between fold change and p-values in a volcano plot, to emphasize that many proteins change only marginally and with a relatively low confidence level. Even then, it may not be surprising that '60% of these of these proteins were not previously identified as regulated by reduced IIS' (page 6), and the claim that 'substantial, new information was gained' remains a leap of faith.

Author response

We agree that more detail was needed to clarify our protein quantification methods and the quality of our proteomic dataset. We have now included these details as explained below.

*The reviewer points out that we have not mentioned the number of biological replicates. Our study is based on 6 biological replicates, across 4 genotypes, and 4 tissues. The **Peptide Generation** section of the methods (pg. 22) now includes the biological replicate number and the text is shown below.*

“Peptide Generation

Fly tissues (50/sample) from 6 biological replicates were lysed in pre-heated (95°C) 6M Guanidinium hydrochloride, 10 mM TCEP, 40 mM CAA, 100 mM Tris pH 8,5 lysis buffer.

Compared to other similar proteomic studies previously published in Molecular Systems Biology and elsewhere, the number of replicates is high. Although such studies do not exist in Drosophila, outside of methodological studies such as the Aradska et al. study (Aradska et al, 2015) described above, studies in C. elegans and rats routinely analyze only up to 3 biological replicates (Stout et al, 2013; Narayan et al, 2016; Ori et al, 2015).

As suggested by the reviewer, we have included additional analyses to show the quality and reproducibility of our dataset. First, we have added supplemental Fig EV1A, a correlation heatmap across all samples to show the similarity between samples. This analysis shows high levels of heterogeneity between tissues. However, within tissues, samples are highly homogeneous with Pearson's correlation coefficient values ranging between 0.90 and 0.98. This level of homogeneity between samples is one benefit of using organisms such as Drosophila, where genetic and environmental conditions can be carefully controlled, thus minimising inter-sample variation, as stated in the methods section (pg. 21).

The combination of a high number of biological replicates and sample homogeneity allows the detection of small magnitude, but significant changes in protein levels in our dataset. As suggested by the reviewer, we have added the following sentence to pg. 6 of the manuscript, making it explicit that many changes we detect are of small magnitude. “In total, expression of 2372 proteins was significantly altered upon reduced IIS (10% FDR, Fig. 2 & Dataset EV4). Out of those 982 of which showed absolute fold changes larger than 2 in at least one tissue”.

As suggested by the reviewer we have generated tissue-specific plots to show the association between log fold change and confidence (volcano plots) and between log fold change and average expression level (MA-plots). Both the MA- and volcano plots have been included in Fig EV1 as sub panels B and C. MA-plots show, as expected from expression data, that on average we can detect changes of smaller magnitude among the more abundant proteins. However, as shown by the volcano plots, we can identify those small magnitude changes with high confidence. For the reviewer and editor we compared the log fold magnitude range of significantly regulated proteins across all tissues at different FDR levels. At an FDR of 0.1 the log fold magnitude range of significantly regulated proteins was 0.18 to 6.1-log fold change, at an FDR of 0.05 it was 0.23 to 5.4-log fold change. Together these additional analyses confirm the robustness and quality of our dataset and emphasise the substantial new information gained.

Point 2. On a technical note, protein identification in this study seems to heavily rely on the 'match between runs' option in MaxQuant. In fact it is unclear why it was used here considering the profound differences in proteome composition between tissues - a scenario for which this option was explicitly NOT designed. Since this introduces the risk of introducing erroneous transfer of peptide identifications, the authors should indicate the FDR of protein assignment between disparate tissues.

Author response

Our analysis was performed with the correct settings. The tissues were run individually through MaxQuant, thus 'matching between runs' only occurred within a tissue. This was not clear from our original text. We have clarified this and the sentence 'Protein and peptide identifications were performed within, but not across tissue groups' has been added to methods section on pg. 24. A summary of the files used and the parameters of each MaxQuant run are available in the PRIDE repository upload (parameters.txt and summary.txt respectively).

As stated in the methods section 'protein identification and quantification', the proteins and peptide identification was performed with a false discovery rate (FDR) of 0.01, which is standard in the field (Aradska et al, 2015; Narayan et al, 2016).

Point 3. Page 7: by comparing proteomic data sets obtained from both biological model systems, the authors identify 361 and 196 proteins whose expression do and do not depend on Foxo, respectively. However, in light of point 1 above, since the number of proteins deemed to change in expression is likely to be inflated, the robustness of this analysis can be questioned. Details should be added on the effect size in protein expression, and the error in these values taking into account error propagation from the respective proteomic data that are compared.

Author response

We draw attention to our response to Reviewer 2's first point. In that response, and now in the manuscript, we have demonstrated the quality and robustness of our dataset. We therefore conclude that the number of differentially regulated proteins is not inflated. To further clarify this point and to show the effect size in protein expression, we have included MA plots in Fig. S1B. These plots show that, as with all expression data, on average we detect more abundant proteins as differentially expressed, however, we also detect many low abundant proteins as differentially regulated.

Reviewer #3:

Point 1. The authors rightly claim that the lifespan-relevant changes should be occurring downstream of FOXO. Since FOXO is a transcription factor, it affects mRNA levels. Yet the effects described here are occurring at the protein level. Hence there is a disconnect between the FOXO-dependent transcriptional changes and the proteome changes. The authors should test whether the main proteins discussed in this manuscript, which change in abundance upon reduced IIS, are FOXO targets. Are the 6 mitochondrial proteins shown in Figure 4A, which increase in the fat body, also up-regulated at the mRNA level in a foxo dependent manner? ie are these bona-fide foxo targets or do they change in abundance as a secondary consequence of changes in foxo targets? Likewise, the mRNA levels for the proteasomal subunits highlighted in yellow in Fig 5A should be quantified and presented (especially Rpt6).

Author response

As stated in our main text, "Long-lived IIS mutants show a major and tissue-specific rearrangement of RNA transcript expression, as a consequence of alteration of the activity of target transcription factors". However, it is clear that whatever the mechanisms acting between FOXO and the phenotype, they are mediated at least in part by proteins. It is also true that some differentially regulated proteins will be direct dFoxo targets and others will be downstream of dFoxo but regulated by other mechanisms such as protein stability and translation, which will still be dependent on dFoxo. Furthermore, it is important to note that those genes/proteins would be missed in a pure RNA seq/ChIP-seq analysis.

We agree with the reviewer that determining if the main proteins in our study are 'bona-fide' dfoxo targets is of interest. However, simply measuring mRNA transcript levels by RTqPCR is not

sufficient to identify a gene as direct Foxo target gene. We have therefore used MEME-predicted *dfoxo* binding sites and results from a previously published *dfoxo* ChIP-seq study (Bai et al, 2013) and compared them to our IIS-regulated proteome datasets.

In total, MEME predicted 2618 genes to have a *dfoxo*-binding site within 1Kb of their transcriptional start site. Furthermore, we quantified 1046 proteins corresponding to genes identified as *foxo*-binding in the study by Bai et al. (Bai et al, 2013) Both sets overlap significantly with proteins responding to reduced IIS (45% and 19% of those, respectively). The overlap with ChIP-seq identified targets is further enriched and significant for *foxo*-dependent proteins (24%).

We have added this analysis to the MS on pg. 7, added data to Dataset EV4, the analysis description in the methods section on pg. 25, and discussed the outcome of the analysis on pg. 17. The edited text is also shown below.

pg.7 “To assess if those 361 proteins were more likely to be direct or indirect targets of *dfoxo*, we searched for predicted *dfoxo* binding motifs within 1kb of their transcriptional start sites using MEME (Bailey et al, 2009). 45% of the 361 proteins came from genes with *dfoxo*-binding motifs (Dataset EV4), and may therefore be directly regulated by dFoxo. However, the remaining 55% of those 361 proteins came from genes lacking *dfoxo*-binding motifs which suggests that although their expression was *dfoxo*-dependent they were not directly regulated transcriptionally by dFoxo.”

pg.25 **“Binding site identification**

We used FIMO, MEME suite, (Bailey et al, 2009) to identify genes whose transcription start site was within 1000 bp of a *foxo* binding motif, using the default p-value threshold ($1e^{-4}$). Sequences corresponding to genes of measured proteins were extracted from the BDGP6 reference genome, while the Foxo binding motif was taken from Fly Factor Survey (Zhu et al, 2011). We then mapped FIMO hits to proteins whose expression changed under reduced IIS conditions or those that were identified as *dfoxo*-dependent and used a hypergeometric test to evaluate the significance of the overlap. Selection and background were limited to the set of genes corresponding to measured proteins.”

pg.17 “Our dataset has thus identified possible novel mediators of responses to reduced IIS. We have also characterized the *dfoxo*-dependent proteomes, and determined which individual protein coding genes contain predicted *dfoxo*-binding motifs, identifying possible direct and indirect targets of *dfoxo*. Importantly, we have thus identified the potentially longevity-associated proteome and the processes it regulates. “

As requested by the reviewer we have also performed RTqPCR analysis of several mitochondrial ETC, and proteasomal subunit genes (Responses figure 3). We do not detect any transcriptional regulation of proteasomal subunits in the guts of mNSC-ablated flies, and no dfoxo-dependent regulation in the case of Rpn6 and Rpt6R. This suggests that the regulation we see on the proteomic level is post-transcriptional, and would only have been detected by analysis of proteins. In response to reduced IIS all mitochondrial ETC genes tested showed increased expression. However, with the exception of CG2014, the level of direct dfoxo-dependency was not clear, again suggesting some degree of post-transcriptional control.

Response Figure 3. Tissue-specific RTqPCR analysis of mitochondrial ETC (fat body-specific) and proteasomal subunits (gut-specific)

Point 2. In Fig 4B, the quantification does not seem to correspond to the image of the western blot. In the WB, NDUF53 protein levels appear reduced in the *foxo94* lysates compared to *wDah* (despite higher loading control levels) yet the quantification does not show this. If NDUF53 is not the best protein to show these changes, then another protein should be shown. In any case, the changes in respiration are unlikely to be due to only one protein changing in abundance.

Author response

The blot quantification shown in Fig 4B is the average NDUF53 levels of four biological replicates. We agree with the reviewer that in the example image there appears to be reduced NDUF53 levels in the *dfoxo⁹⁴* lysates. This reduction was not present in the other biological replicates. We have therefore exchanged the western blot image in Fig 4B to one that is more representative. To verify our analysis of the blots for the reviewer, we have shown all biological replicates in a figure below (Responses Figure 4).

Responses Figure 4. Western blot analysis of NUDFS3 levels in the fat body of control (wDah), dfoxo null flies (dfoxo⁹⁴), mNSC-ablated flies (InsP3-Gal4/UAS-rpr), and mNSC-ablated flies lacking dfoxo (InsP3-Gal4/UAS-rpr, dfoxo⁹⁴). Individual biological replicates are shown.

Point 3. In Figure 4F, are spargel protein and mRNA levels in the fat body actually affected by IIS-ablation and are those changes foxo-dependent? If not, this should be clearly stated. Otherwise the text gives the impression that regulation of mitochondrial activity in response to reduced IIS goes via Spargel regulation.

Author response

We did not detect either spargel or delg in adult fat body. This is unsurprising because, according to publicly available RNA-seq data from ENCODE (Graveley et al, 2011), both spargel and delg are only weakly expressed. We thank the reviewer for pointing this out and we have now edited the text to clarify this point. "Spargel and delg are both expressed at low levels in adult Drosophila fat body (Graveley et al, 2011), and we did not detect them, or their regulation in our analysis (Dataset EV1). However, as with many TFs and cofactors, the activity of Spargel and delg could be regulated on the posttranslational level (Li et al, 2007)."

Point 4. Are the changes in Rpt6 levels shown in Figure 5B also occurring in a foxo mutant background? (ie are they foxo dependent?). This should be tested at the mRNA and protein levels.

Author response

Rpt6 protein levels are dfoxo-dependently increased in the gut of mNSCs-ablated flies. This information, in the form of a representative western blot and corresponding quantification, has now been included in Figure 5B and in the main text on pg. 11 "In our analysis, the proteasomal subunit Rpt6 showed the greatest degree of regulation, increasing 2.6 fold in the gut. We confirmed this, and that this change was dfoxo-dependent, by western blot analysis of the guts of mNSC-ablated flies compared to controls (Fig. 5B)." Rpt6R mRNA levels were tested and described in response to point #1.

Point 5. Regarding Figure 5, the authors conclude: "These data suggest that reduced IIS results in increased proteasomal assembly in the gut, through increased levels of Rpt6". There do not seem to be any data showing that the increased proteasomal assembly is due to increased levels of Rpt6. Only correlative data are shown. Either data should be provided, or the claim should be fixed.

Author response

We agree with the reviewer. We do not provide direct evidence that the increase in proteasomal assembly is mediated by Rpt6. We have removed this claim in the main text. The text now states 'These data suggest that reduced IIS results in increased proteasomal assembly in the gut, possibly through increased levels of Rpt6'.

Minor point 1. Have the authors tested if the discovered changes in translation rates in the fat body are linked to lifespan? This should be mentioned/discussed in the text.

Author response

These experiments have not been performed and are thus not discussed in the main text.

Minor point 2. Figure S2: With the exception of the fat body, the other organs show a very high variability in the S35-incorporation and this does not seem to correspond to the size of the error bars in the quantifications. What are the error bars showing? The figure legends do not indicate if the error bars are std. dev. and over how many biological replicates. This information should be added, and perhaps the error bars re-calculated.

Author response

The error bars show Standard error of the mean. This is now stated in the figure legend. We have also clarified the number of replicates used in this experiment. The clarified figure legend (Fig EV3) is shown below.

“Figure EV3 Translational activity in the fat body under reduced IIS is reduced in a dfoxo-dependent manner. (A) De-novo protein synthesis as measured by incorporation of 35S into proteins from ex vivo fat body, (B) head, (C) thorax, and (D) gut tissue. Equal levels of protein were loaded per lane. Representative gel exposures show fat body 35S incorporation from w^{Dah} , $dfoxo^{A94}$, $InsP3-Gal4/UAS-rpr$, and $InsP3-Gal4/UAS-rpr, dfoxo^{A94}$ flies. Quantification of gel exposures normalized to total protein and corresponding two-way ANOVA, as well as Bonferroni corrected post hoc tests are shown alongside each gel exposure, (Error bars show SEM, $n=11$ for w^{Dah} and $InsP3-Gal4/UAS-rpr$ fat body samples, for all other genotypes and tissues $n=5$, $*=p<0.05$, $***=p<0.001$).”

Minor point 3. Figure 4C-D: the respiration assays are missing from the materials & methods.

Author response

The respiration rate assay methodology has been included in the methods section, in the subsection 'Respiratory rate measurements'. The updated methods section is shown below.

Respiratory rate measurements

Aged flies (10d) were dissected in PBS and transferred to respiratory buffer (120 mM sucrose, 50 mM KCl, 20 mM Tris-HCl, 4 mM KH₂PO₄, 2 mM MgCl₂, 1 mM EGTA, 0.01% digitonin, 0.05% BSA, pH 7.2). Oxygen consumption was measured using an oxygraph chamber (OROBOROS) at 25°C. Complex I-dependent respiration was assessed using the substrates proline (10mM), pyruvate (10mM), malate (5mM) and glutamate (5mM), along with ADP (1.25mM). The respiration was uncoupled by the addition of CCCP (0.3uM). Maximum flux was then measured by adding complex II substrates succinate (10mM) and glycerol-3-phosphate (5mM). Rotenone-sensitive flux was measured in the presence of rotenone (3uM). Finally, dry weights of tissues were determined to normalise the oxygen consumption.

Minor point 4. The authors write "Furthermore, expression of Spargel or delg in the fat body of dilp2-3,5 mutants was required for longevity but did not affect the lifespan of wild type controls (Fig. 4F, S3D-E) " but this conclusion seems to strong given the provided data. Spargel is not

required for the increase in lifespan upon IIS-ablation, but it is contributing to it as the lifespan of Spargel knockdown animals still can be significantly prolonged by IIS-ablation.

Author response

We have edited the text to incorporate the advice of the reviewer and reduced the strength of the statement. Furthermore, knock-down of expression of Spargel or delg in the fat body of dilp2-3,5 mutants reduced the extent of their increased longevity, but did not affect the lifespan of wild type controls (Fig. 4F, EV4D-E).

Minor point 5. Figure 6I - The lifespan effect is very minor. Rephrase as "small but significant"?

Author response

The text has now been rephrased as requested by the reviewer. "We independently confirmed this finding by over-expressing RPN6 in the adult gut using the inducible, gut-specific GeneSwitch driver TIGS-2, which was sufficient to increase proteasome activity, increase gut integrity and induce a small, but significant, extension of lifespan (Fig 6I, Fig EV6A-C)."

Minor point 6. Wrong figure reference: "Adult-specific over-expression of RPN6 was sufficient to increase proteasome activity, increase gut integrity and extend lifespan (Fig 6I, Fig S4A-C)." Should be "(Fig 6I, Fig S5A-C)" ?

Author response

We thank the reviewer for finding this error, which has now been corrected in the main text.

Minor point 7. The protein expression data will be of great value to the community. The authors should summarize in a supplemental table (excel? csv?) these data, showing protein expression values (or peptide counts?) in the four tissues for the various genotypes (+/- IIS, +/- FOXO).

Author response

We agree with the reviewer that our protein expression data will be of great value to the scientific community. Firstly, we have deposited our raw data on the PRIDE repository, and on publication these data will become freely available. Secondly, we have done as the reviewer suggested and generated an Excel file containing average expression LFQ values, and SEM, for all tissues and for all genotypes. We also make a point of referring to this resource in the main text on pg. 5 and 16 as Dataset EV1

Response letter reference list

- Aradska J, Bulat T, Sialana FJ, Birner-Gruenberger R, Erich B & Lubec G (2015) Gel-free mass spectrometry analysis of *Drosophila melanogaster* heads. *Proteomics* **15**: 3356–3360
- Bai H, Kang P, Hernandez AM & Tatar M (2013) Activin signaling targeted by insulin/dFOXO regulates aging and muscle proteostasis in *Drosophila*. *Plos Genet* **9**: e1003941
- Bailey TL, Boden M, Buske FA, Frith M, Grant CE, Clementi L, Ren J, Li WW & Noble WS (2009) MEME SUITE: tools for motif discovery and searching. *Nucleic Acids Research* **37**: W202–8
- Graveley BR, May G, Brooks AN, Carlson JW, Cherbas L, Davis CA, Duff M, Eads B, Landolin J, Sandler J, Wan KH, Andrews J, Brenner SE, Cherbas P, Gingeras TR, Hoskins R, Kaufman T & Celniker SE (2011) The *D. melanogaster* transcriptome: modENCODE RNA-Seq data for dissected tissues.
- Li X, Monks B, Ge Q & Birnbaum MJ (2007) Akt/PKB regulates hepatic metabolism by directly inhibiting PGC-1 α transcription coactivator. *Nature* **447**: 1012–1016
- Murali T, Pacifico S, Yu J, Guest S, Roberts GG & Finley RL (2011) DroID 2011: a comprehensive, integrated resource for protein, transcription factor, RNA and gene interactions for *Drosophila*. *Nucleic Acids Research* **39**: D736–43
- Narayan V, Ly T, Pourkarimi E, Murillo AB, Gartner A, Lamond AI & Kenyon C (2016) Deep Proteome Analysis Identifies Age-Related Processes in *C. elegans*. *Cell Syst*: 1–35
- Ori A, Toyama BH, Harris MS, Bock T, Iskar M, Bork P, Ingolia NT, Hetzer MW & Beck M (2015) Integrated Transcriptome and Proteome Analyses Reveal Organ-Specific Proteome Deterioration in Old Rats. *Cell Syst* **1**: 224–237

Stout GJ, Stigter ECA, Essers PB, Mulder KW, Kolkman A, Snijders DS, van den Broek NJF, Betist MC, Korswagen HC, MacInnes AW & Brenkman AB (2013) Insulin/IGF-1-mediated longevity is marked by reduced protein metabolism. *Mol Syst Biol* **9**: 679–679

Vanunu O, Magger O, Ruppin E, Shlomi T & Sharan R (2010) Associating genes and protein complexes with disease via network propagation. *PLoS Comput. Biol.* **6**: e1000641

Zhu LJ, Christensen RG, Kazemian M, Hull CJ, Enameh MS, Basciotta MD, Brasfield JA, Zhu C, Asriyan Y, Lapointe DS, Sinha S, Wolfe SA & Brodsky MH (2011) FlyFactorSurvey: a database of *Drosophila* transcription factor binding specificities determined using the bacterial one-hybrid system. *Nucleic Acids Research* **39**: D111–7

2nd Editorial Decision

09 August 2017

Thank you for sending us your revised manuscript. We have now heard back from the two referees who were asked to evaluate your study. As you will see below, the reviewers think that most of the issues raised in the first round of review have now been satisfactorily addressed. However, reviewer #3 raises two remaining issues, which we would ask you to address in a minor revision.

REVIEWER REPORTS

Reviewer #2:

The authors have appropriately addressed my concerns. This has rendered this impressive study even more insightful, so I recommend acceptance for publication.

Reviewer #3:

The authors have done a good job of addressing the concerns raised in the original review.

That said, in response to my Main Issue #1, the authors provide a reply to my specific question in the rebuttal letter as Response Figure 3, but these data are not incorporated into the figure. These data essentially show that the main changes at the proteomic level that are the focus of this manuscript are not due to direct transcriptional effects of FOXO. This is fine, because the whole purpose of such a proteomic study is to pick up effects at the protein level that would otherwise go missed at the mRNA level due to post-transcriptional and post-translational effects. However, the most obvious link between FOXO and these genes would be a direct transcriptional effect, which is not the case. So these data should not be hidden from the reader. These data show that something more complex is going on, which might be a starting point for future studies by the community. Hence the data shown in Response Figure 3 need to be incorporated into the manuscript and discussed.

Furthermore, the loading control of Fig 5B is of very poor quality.

2nd Revision - authors' response

15 August 2017

Reviewer #3 comment

The authors have done a good job of addressing the concerns raised in the original review.

That said, in response to my Main Issue #1, the authors provide a reply to my specific question in the rebuttal letter as Response Figure 3, but these data are not incorporated into the figure. These data essentially show that the main changes at the proteomic level that are the focus of this manuscript are not due to direct transcriptional effects of FOXO. This is fine, because the whole purpose of such a proteomic study is to pick up effects at the protein level that would otherwise go missed at the mRNA level due to post-transcriptional and post-translational effects. However, the most obvious link between FOXO and these genes would be a direct transcriptional effect, which is

not the case. So these data should not be hidden from the reader. These data show that something more complex is going on, which might be a starting point for future studies by the community. Hence the data shown in Response Figure 3 need to be incorporated into the manuscript and discussed.

We agree with Reviewer #3 and have now included the data shown in Response Figure 3 into the manuscript as Figure EV7 A-C and it is discussed in the final paragraph of pg 16. The edited text is shown below and underlined

Our dataset also identified possible novel mediators of responses to reduced IIS. For example, our proteomic analysis, suggested a gut specific regulation of proteasomal function in response to reduced IIS, which was confirmed by finding a corresponding gut-specific proteasomal phenotype. We have also characterized dfoxo-dependent changes in protein expression in IIS mutant flies, separating those changes associated with IIS-mediated longevity from those changes associated to other IIS-mediated phenotypes. Furthermore, we determined which IIS responsive protein coding genes contain predicted dfoxo-binding motifs, identifying possible direct and indirect targets of dfoxo. Additionally, we examined transcriptional changes in several regulated candidate genes associated to mitochondria and the proteasome (Fig. EV7A-C). Some candidate genes were regulated in a dfoxo-dependent manner, consistent with a direct regulation by dfoxo, however many did not reflect the changes seen at the protein level, suggesting indirect regulation by dfoxo, possibly through post-transcriptional and/or post-translational regulation (Fig. EV7A-C). Importantly, we have thus identified longevity-associated changes in the proteome of IIS mutant flies and the processes that they regulate, and we have experimentally demonstrated the role of some of these processes in extension of lifespan.

The corresponding figure legend for Fig. EV7A-C has also been included on pg. 36 and the appropriate method included in the methods section on pg. 30. Both are shown below.

Figure EV7. Q-RT-PCR analysis of tissue-specific candidate gene expression in response to reduced IIS. A-C. Q-RT-PCR analysis shows that of CG2014, CG6463, CoIV, CG4169, CG3731 in the fat body (A), or of Rpn6, Rpt6R, Rpn11, and Pros-alpha3 in gut (B-C) of control (w^{Dah}), dfoxo mutant ($dfoxo^{94}$) mNSC-ablated ($InsP3-Gal4/UAS-rpr$), and mNSC-ablated in the absence of dfoxo ($InsP3-Gal4/UAS-rpr$, $dfoxo^{94}$). Relative expression levels are normalized to either RPL32 (A) and Actin5c (B-C). Significance established by two-way ANOVA and post hoc pairwise tests ($n=3$, or otherwise shown). Bars indicate mean SEM (* $p<0.05$; ** $p<0.01$; *** $p<0.001$; **** $p<0.0001$).

Quantitative real-time PCR

Total RNA was extracted using Trizol (Invitrogen Corp.) according to the manufacturer's instructions, including a DNase treatment. cDNA was prepared using SuperScript III first strand synthesis kit (Invitrogen Corp.). Quantitative real-time PCR was performed in a 7900HT real-time PCR system (Applied Biosystems). Relative expression (fold induction) was calculated using the $\Delta\Delta C_T$ method and Rpl32 or Actin5c as a normalization control.

The loading control of Fig 5B is of very poor quality

We have replaced the loading control image with a better quality, shorter exposure image.

Thank you again for sending us your revised manuscript. We are now satisfied with the modifications made and I am pleased to inform you that your paper has been accepted for publication.

Corresponding Author Name: Linda Partridge and Andreas Beyer

Manuscript Number: MSB-17-7663